# Communication-Efficient Federated Low-Rank Update Algorithm and its Connection to Implicit Regularization

## Abstract

Federated Learning (FL) faces significant challenges related to communication efficiency and heterogeneity. To address these issues, we explore the potential of using low-rank updates. Our theoretical analysis reveals that client's loss exhibits a higher rank structure (gradients span higher rank subspaces of Hessian) compared to the server's loss. Based on this insight, we hypothesize that constraining client-side optimization to a low-rank subspace could provide an implicit regularization effect. Consequently, we propose FedLoRU, a general low-rank update framework for FL. Our framework enforces low-rank client-side updates and accumulates these updates to form a higher-rank model. Additionally, variants of FedLoRU can adapt to environments with statistical and model heterogeneity by employing multiple or hierarchical low-rank updates. Experimental results demonstrate that FedLoRU performs comparably to full-rank algorithms and exhibits robustness to heterogeneous and large numbers of clients.

## 1 Introduction

Federated learning (FL, (McMahan et al., 2017)) is a collaborative learning framework designed to enhance privacy preservation in machine learning applications. This approach has gained importance due to rising concerns over data privacy, as it allows multiple participants to train a model collectively without sharing raw data.

While FL offers privacy benefits, it trades off some performance compared to centralized learning. Two primary factors contributing to this trade-off are communication overhead and heterogeneity. Despite improvements in computation and memory capacities, communication speeds have only slightly improved, making communication overhead a major factor in slowing down FL (Zheng et al., 2020). Additionally, various forms of heterogeneity—statistical, system, and device—further complicate FL (Ye et al., 2023; Kairouz et al., 2021). These issues are especially pronounced with a large number of clients, where frequent, less impactful updates slow down training and reduce performance.

Addressing these challenges is becoming increasingly critical, for example, training large language models (LLMs) in an FL framework. Utilizing private datasets on edge devices for LLM training is promising due to the limited availability of public data (Ye et al., 2024). However, this approach presents significant issues, notably in terms of communication overhead, as edge devices possess heterogeneous resources and data. Additionally, the need for effective regularization across clients is required. Consequently, the development of algorithms to tackle these challenges is an essential problem to bridge the gap between practical and conceptual FL applications.

There has been substantial research focusing on the low-rank characteristics in centralized learning. By low rank, we refer to gradients spanning a low rank subspace of Hessian at any given weights or the weight matrix being of the form $AB$ where the number of columns of $A$ is low. By utilizing low-rank factorized update models such as LoRA (Hu et al., 2021), DyLoRA (Valipour et al., 2022), and QLoRA (Dettmers et al., 2024), the number of trainable parameters can be reduced, which helps conserve memory and computational resources. Further observations (Huh et al., 2021; Ji & Telgarsky, 2018) indicate that over-parameterized models tend to find low-rank solutions, which provide implicit regularization effects.

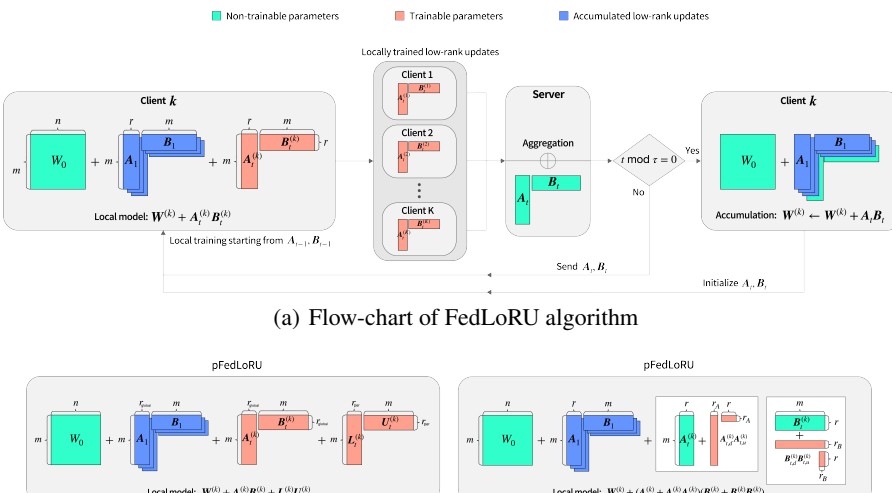

(a) Flow-chart of FedLoRU algorithm

(b) Low-rank factorization methods in pFedLoRU and mFedLoRU

Figure 1: Figure 1(a) provides a flowchart representing the FedLoRU algorithm. In this algorithm, the model training is conducted solely using rank-$r$ matrices, with communication between the server and clients being confined to these matrices. Clients incrementally add low-rank update matrices to their local base model $W^{(k)}$ every $\tau$ rounds, resulting in a higher-rank model. Figure 1(b) depicts the utilization of low-rank factorization within the pFedLoRU and mFedLoRU algorithms. For clarity, the equations assume all $\alpha$ parameters are set to 1.

However, the rank properties of the loss landscape in FL remain under-explored. Here, we first analyze the difference in the stable rank—defined as the squared ratio of the Frobenius norm to the spectral norm—between client Hessians and the server Hessian of any weights, discovering that client exhibits a higher-rank structure. Based on this theoretical insight, we hypothesize that the higher-rank structure of client's loss contributes to increased client discrepancy and that restricting client-side updates could provide an implicit regularization effect across clients. This leads us to the research question:

*Can we use low-rank updates in federated learning to achieve both communication overhead reduction and regularization effects across clients?*

We propose the Federated Low-Rank Updates (FedLoRU) algorithm, which addresses communication overhead and the challenges posed by a large number of clients by employing client-side low-rank updates and server-side accumulation of low-rank updates. FedLoRU factorizes client-side update matrices $A$ and $B$ and applies iterative optimization to these low-rank factorized matrices. Clients and the server share the factorized matrices, which the server then aggregates. Matrices $A$ and $B$ are being communicated between the clients and server, rather than the much larger matrix $AB$. To make the model's weight rank high, the server successively accumulates low-rank matrices. We also generalize the low-rank update strategy within federated learning for various heterogeneous settings.

Our comprehensive approach underscores the potential of low-rank updates not only to enhance communication efficiency but also to impose implicit regularization and harmonize the optimization process across heterogeneous federated learning settings. Our contributions can be summarized as follows. 1) We propose FedLoRU, the first algorithm using successive low-rank updates for both pre-training and fine-tuning in federated learning, and introduce variants of FedLoRU for personalization and model heterogeneity settings; 2) We investigate the rank properties of client and server losses, analytically showing that under stochastic sampling, the rank of the Hessian of the loss function increases with smaller sample sizes; 3) We provide empirical evidence of the higher rank structure of client losses and demonstrate that restricting the rank of local updates aids in implicit regularization; 4) On average, FedLoRU improves state-of-the-art communication-efficient federated learning algorithms on a variety of datasets, including LLM fine-tuning, and exhibits superior performance as the number of clients increases.

## 2 RELATED WORK

**Communication-Efficient Federated Learning**  Extensive research has addressed communication challenges in FL (Shahid et al., 2021). FedPAQ (Reisizadeh et al., 2020) and AdaQuantFL (Jhunjhunwala et al., 2021) employ quantization to reduce the precision of weights, while Fed-Dropout (Caldas et al., 2018) and FedMP (Jiang et al., 2023) apply pruning to remove less important weights. Since quantization and sparsification do not alter the core network structure, they can be easily combined with other algorithms (e.g., FedLoRU) to reduce communication overhead.

In contrast, model compression techniques modify the model structure itself by compressing the original model before communication and restoring it afterward. FedDLR (Qiao et al., 2021) compresses using low-rank approximation for both server-to-client and client-to-server communication but reverts to the full model for local training. FedHM (Yao et al., 2021) compresses only during server-to-client communication, where clients train factorized low-rank models that are aggregated by the server. Although both methods reduce communication overhead, their server-side compression approaches can lead to performance degradation. To mitigate potential information loss during server-side compression, we focus on client-side factorization, avoiding compression processes.

**Low-rank nature of centralized and federated learning**  Numerous studies (Gur-Ari et al., 2018; Li et al., 2018; Sagun et al., 2016) assert that the training process in deep learning inherently possesses a low-rank nature. Low-Rank Adaptation (LoRA, Hu et al. (2021)) is a representative algorithm that leverages this low-rank characteristic, particularly for fine-tuning tasks, by freezing pre-trained weights and applying low-rank updates via the decomposition $W = W_0 + AB$, where $W_0 \in \mathbb{R}^{m \times n}$, $A \in \mathbb{R}^{m \times r}$, $B \in \mathbb{R}^{r \times n}$, $r \ll m, n$. However, effectively leveraging the low-rank structure during pre-training remains a challenge, as the weights do not inherently exhibit a low-rank nature (Yu & Wu, 2023; Zhao et al., 2024). To address this, ReLoRA (Lialin et al., 2023) seeks to achieve a higher-rank model by accumulating multiple low-rank updates, expressed as $W = W_0 + \sum_{i=1}^{M} A_i B_i$ where $A_i \in \mathbb{R}^{m \times r}$, $B_i \in \mathbb{R}^{r \times n}$.

In federated learning, some research has aimed to exploit the low-rank nature observed in centralized learning. LBGM (Azam et al., 2021) and FedLRGD (Jadbabaie et al., 2023) approximate gradients using past or sampled gradients, assuming gradients lie in a low-rank subspace. However, there is a noticeable gap in analyzing rank characteristics specific to federated learning. In the context of federated learning, there is a complex loss landscape involving multiple client-side and a single server-side optimization, and leveraging a low-rank structure needs to consider their respective rank structures. To our knowledge, no prior work has examined the rank structure in federated learning contexts without making very stringent assumptions. Our study is pioneering in addressing this gap, using analytical results and insights to develop a novel algorithm.

**Low-Rank Adaptation in Federated Learning**  Recent studies have studied the application of LoRA within federated learning frameworks. Notable algorithms, such as FedLoRA (Wu et al., 2024; Yi et al., 2023), FFALoRA (Sun et al., 2024), and Hyperflora (Lu et al., 2024), employ LoRA adapters to facilitate personalization. These methods apply low-rank adaptation to a pre-trained model during the local personalization training phase. On the other hand, other works (Zhang et al., 2023; Kuo et al., 2024; Cho et al., 2023) apply LoRA for fine-tuning within federated learning environments.

These approaches use only one low-rank matrix that restricts the model to a low-rank subspace. In contrast, we utilize multiple accumulated low-rank matrices allowing the model to achieve higher rank. Specifically, we extend the concept of LoRA by incorporating client-side low-rank updates and server-side accumulation to address the low-rank limitation of LoRA as well as the challenges posed by communication and client-server rank disparity. We also generalize the low-rank strategy within federated learning for both pre-training and fine-tuning, and for heterogeneous environments.

## 3 LOW-RANK UPDATES IN FEDERATED LEARNING

In centralized learning, neural network losses exhibit a low-rank structure, indicating that the gradient lies within the subspace spanned by the $k$ eigenvectors of the Hessian during training (Gur-Ari et al., 2018). While efforts have been made to utilize this low-rank structure to enhance federated

learning algorithms, there is a lack of studies analyzing the rank structure of federated learning. In federated learning, the clients and server have distinct losses, resulting in different rank structures. Understanding these differing rank structures of client and server losses is crucial for developing low-rank-inspired algorithms tailored for federated learning.

In this section, we theoretically analyze the rank structures in federated learning, particularly comparing the stable rank of client and server Hessians. Based on this analysis, we propose FedLoRU, a novel federated learning algorithm aimed at improving communication efficiency and addressing performance degradation with a large number of clients. We also present a variant of FedLoRU to handle model and statistical heterogeneity in federated learning.

### 3.1 Higher rank nature of clients in federated learning

**Notation and problem setup**   Suppose $\psi(\boldsymbol{x}, \boldsymbol{y})$ is a data generating distribution for an input-output pair $(\boldsymbol{x}, \boldsymbol{y}) \in \mathbb{R}^{d_x} \times \mathbb{R}^{d_y}$. We consider the problem of finding a prediction function $h^R(\cdot; \cdot)$ : $\mathbb{R}^{d_x} \times \mathbb{R}^R \to \mathbb{R}^{d_y}$ parameterized by a $R$-dim weight vector $\omega^R \in \mathbb{R}^R$. Given a loss function $\ell(\cdot, \cdot)$ : $\mathbb{R}^{d_y} \times \mathbb{R}^{d_y} \to \mathbb{R}$, the true risk $\mathcal{L}_{\text{true}}(h^R, \omega^R) = \int \ell(h^R(\boldsymbol{x}; \omega^R), \boldsymbol{y}) d\psi(\boldsymbol{x}, \boldsymbol{y})$ is defined as the loss over the data-generating distribution $\psi(\boldsymbol{x}, \boldsymbol{y})$. The corresponding true Hessian is $\boldsymbol{H}_{\text{true}}(h^R, \omega^R) = \nabla^2 \mathcal{L}_{\text{true}}(h^R, \omega^R)$. If $\mathcal{D}_N = \{(\boldsymbol{x}_1, \boldsymbol{y}_1), \cdots, (\boldsymbol{x}_N, \boldsymbol{y}_N)\}$ is a dataset generated from the distribution $\psi$, the empirical loss and Hessian for $\mathcal{D}_N$ are $f_N(h^R, \omega^R) = \sum_{(x,y) \in \mathcal{D}_N} \frac{1}{N} \ell(h^R(x; \omega^R), y)$ and $\boldsymbol{H}_N(h^R, \omega^R) = \sum_{(x,y) \in \mathcal{D}_N} \frac{1}{N} \frac{\partial^2}{\partial (\omega^R)^2} \ell(h^R(x; \omega^R), y)$.

We consider a random selection of $M$ samples without replacement from $\mathcal{D}_N$ to form a sub-dataset $\mathcal{D}_M \subseteq \mathcal{D}_N$. Let $f_M(h^R, \omega^R)$ and $\boldsymbol{H}_M(h^R, \omega^R)$ denote the loss and Hessian for the sub-dataset $\mathcal{D}_M$. In federated learning, $f_N$ can be considered as the loss that the server optimizes, while $f_M$ represents the loss of a local client assuming the homogeneous setting.

For non-zero real numbers $\theta_1, \cdots, \theta_k$, define $\Omega^R(\theta_1, \cdots, \theta_k)$ as the family of pairs $(h^R, \omega^R)$, where $h^R$ is an $R$-dimensional prediction function and $\omega^R$ is a weight vector, such that the true Hessian has eigenvalues $\theta_1, \cdots, \theta_k$. Specifically, $\Omega^R(\theta_1, \cdots, \theta_k) = \{(h^R, \omega^R) : \boldsymbol{H}_{\text{true}}(h^R, \omega^R) \text{ has eigenvalues } \theta_1, \cdots, \theta_k\}$. Let $\Omega(\theta_1, \cdots, \theta_k) = \bigcup_R \Omega^R(\theta_1, \cdots, \theta_k)$, representing the union of $\Omega^R(\theta_1, \cdots, \theta_k)$ over all dimensions $R$. We aim to show that the difference in stable rank between the Hessians of a server and a client eventually becomes positive as dimension $R$ approaches infinity within the space of $\Omega(\theta_1, \cdots, \theta_k)$, which contains infinitely many $R$ for which $\Omega^R(\theta_1, \cdots, \theta_k) \neq \emptyset$, as proved in Appendix A.1.

**Comparing the stable rank of the client and server Hessians**   Now, we will focus on comparing the stable rank of the client and server Hessians. For given $p, q \in \mathbb{N}$, let $\theta_1 > \cdots > \theta_p > 0 > \theta_{p+1} > \cdots > \theta_{p+q}$ be deterministic non-zero real numbers, and let $(h^R, \omega^R) \in \Omega(\theta_1, \cdots, \theta_{p+q})$ for some $R$. To compare the stable rank of Hessians for two datasets $\mathcal{D}_N$ and $\mathcal{D}_N$, we consider the additive perturbed model of the true Hessian as described by Baskerville et al. (2022):

$$\boldsymbol{H}_N(h^R, \omega^R) = \boldsymbol{H}_{\text{true}}(h^R, \omega^R) + \epsilon^R(N), \quad \boldsymbol{H}_M(h^R, \omega^R) = \boldsymbol{H}_{\text{true}}(h^R, \omega^R) + \epsilon^R(M). \quad (1)$$

Here, $\epsilon^R(N), \epsilon^R(M) \in \mathbb{R}^{R \times R}$ are defined as random error matrices associated with each Hessian. These matrices are assumed to be scaled according to $\epsilon^R(N) = s(N)X^R$, where $X^R \in \mathbb{R}^{R \times R}$ is a random real symmetric matrix and $s : \mathbb{N} \to (0, 1)$ is a decreasing function.

Another study (Granziol et al., 2022) employs the model $\boldsymbol{H}_M(h^R, \omega^R) = \boldsymbol{H}_N(h^R, \omega^R) + \epsilon^R$, implying a dependency structure between $\boldsymbol{H}_M$ and $\boldsymbol{H}_N$. However, their analysis assumes independence between these matrices, which is problematic given the underlying model and practical considerations. In contrast, we address this issue by introducing two decoupled additive perturbed models. Additionally, while Granziol et al. (2022) investigates outlier eigenvalues, our focus is on the difference in the rank of the Hessians.

We seek to determine the limiting eigenvalues of the Hessians $\boldsymbol{H}_N(h^R, \omega^R)$ and $\boldsymbol{H}_M(h^R, \omega^R)$ in relation to the eigenvalues of $\boldsymbol{H}_{\text{true}}(h^R, \omega^R)$. Since $(h^R, \omega^R) \in \Omega^R(\theta_1, \cdots, \theta_{p+q})$, the eigenvalues of $\boldsymbol{H}_{\text{true}}(h^R, \omega^R)$ are $\theta_1, \cdots, \theta_{p+q}$. Next, we need to make some assumptions about the random error matrix $X^R$. Assume $X^R$ is a random real symmetric matrix with eigenvalues

$\lambda_1(X^R), \cdots, \lambda_R(X^R)$ and a limiting spectral density $\mu$, such that $\frac{1}{R}\sum_{i=1}^{R}\delta(\lambda - \lambda_i(X^R)) \to \mu(\lambda)$, with convergence in the weak almost sure sense. Examples of matrices exhibiting a well-defined limiting spectral density include Wigner matrices, Wishart matrices, and Gaussian ensembles. We assume $\mu$ is a compactly supported probability measure on $[l_\mu, r_\mu]$ which admits a smooth density with respect to the Lebesque measure and the eigenvectors of $X^R$ obey quantum unique ergodicity (QUE). For more detail about the QUE condition, we refer to Baskerville et al. (2022). We can now find the limiting eigenvalues of $\boldsymbol{H}_N$ and $\boldsymbol{H}_M$.

**Proposition 3.1** (Limiting eigenvalues of $\boldsymbol{H}_N$ (modified from Baskerville et al. (2022))). *Let $R$ be any integer such that $R \geq \bar{R}$ where $\bar{R}$ is the smallest integer such that $\Omega^{\bar{R}}(\theta_1, \cdots, \theta_{p+q})$ is non-empty. For any pair $(h^R, \omega^R) \in \Omega^R(\theta_1, \cdots, \theta_{p+q})$, consider the Hessian additive error model given by $\boldsymbol{H}_N(h^R, \omega^R) = \boldsymbol{H}_{true}(h^R, \omega^R) + \epsilon^R(N)$. If $\lambda_i(\boldsymbol{H}_N(h^R, \omega^R))$ denotes the $i$-th eigenvalue of $\boldsymbol{H}_N(h^R, \omega^R)$, then for $i = 1, \cdots, p$, the following holds:*

$$\lambda_i(\boldsymbol{H}_N(h^R, \omega^R)) \to \begin{cases} g_N^{-1}(\theta_i) & \text{if } g_N^{-1}(\theta_i) > U_N \\ U_N & \text{otherwise} \end{cases} \tag{2}$$

*as $R \to \infty$, and for $i = 0, \cdots, q-1$, we have*

$$\lambda_{R-i}(\boldsymbol{H}_N(h^R, \omega^R)) \to \begin{cases} g_N^{-1}(\theta_{p+q-i}) & \text{if } g_N^{-1}(\theta_{p+q-i}) < L_N \\ L_N & \text{otherwise}. \end{cases} \tag{3}$$

*Here,*

$$g_N^{-1}(\theta) = \theta + s(N)\mathcal{R}_\mu(s(N)\theta^{-1}) \tag{4}$$

*and $U_N$ and $L_N$ are lower and upper bounds of the limiting distribution $\mu_N$ of $\epsilon^R(N)$. In addition, for $p < i \leq P - q$, we have $\lambda_i(\boldsymbol{H}_N(h^R, \omega^R)) \to \{0, L_N, U_N\}$.*

Convergence in Proposition 3.1 is weak almost sure convergence and $\mathcal{R}_\mu(\omega)$, known as the $\mathcal{R}$-transform, is defined by $\mathcal{R}_\mu(t) = S_\mu^{-1}(t) - \frac{1}{t}$ where $S_\mu(t)$ is the Stieltjes transform. Compared to Baskerville et al. (2022), which focuses solely on outlier eigenvalues, we extend the analysis to bulk eigenvalues and adopt a simpler form of $\mu$. Within the proposition, the $i$-th largest or smallest limiting eigenvalues of $\boldsymbol{H}_N$ are determined by the values of $g^{-1}(\theta_i)$. If $g^{-1}(\theta_i)$ falls within the support of the limiting distribution $\mu_N$, the corresponding limiting eigenvalues converge to the bounds. If $g^{-1}(\theta_i)$ does not lie within this support, it converges to $g^{-1}(\theta_i)$ itself; these eigenvalues are typically referred to as outlier eigenvalues in the literature. The detailed proof is provided in Appendix A.2 and is similar to the proof in Baskerville et al. (2022).

**Stable rank**   To compare the rank properties of Hessians of a client and the server, we use the stable rank $\text{srank}(\boldsymbol{A}) = \frac{\|\boldsymbol{A}\|_F^2}{\|\boldsymbol{A}\|_2^2} = \frac{\sum_{i=1}^{n}\sigma_i^2(\boldsymbol{A})}{\sigma_1^2(\boldsymbol{A})}$, which is the square of the ratio between a matrix's Frobenius and spectral norms. Here, $n$ is the rank of matrix $\boldsymbol{A}$, and $\sigma_i(\boldsymbol{A})$ represents its $i$-th singular value. Stable rank serves as a continuous proxy for $\text{rank}(\boldsymbol{A})$ and is known for its robustness against small perturbations. In fact, stable rank—which emphasizes eigenvalues near the top eigenvalue—can be considered a more accurate surrogate of the rank structure of the Hessian considering the empirical evidences that gradients are highly influenced by the top Hessian eigenvector, i.e., the eigenvectors corresponding to the largest eigenvalues. Additionally, bounds on the stable rank of a weight vector provide control over model's complexity (Georgiev et al., 2021).

In the following theorem, we demonstrate that smaller dataset results in a higher limiting stable rank. Furthermore, given that modern neural network models typically possess a very large number of parameters, this finding is applicable to contemporary models.

**Theorem 3.2** (Higher rank nature of Hessian of smaller dataset). *Let $N > M > \bar{R}$ be any integers where $\bar{R}$ is the smallest integer such that $\Omega^{\bar{R}}(\theta_1, \cdots, \theta_{p+q})$ is non-empty. For any pair $(h^R, \omega^R) \in \Omega^R(\theta_1, \cdots, \theta_{p+q})$, let $\boldsymbol{H}_N(h^R, \omega^R)$ and $\boldsymbol{H}_M(h^R, \omega^R)$ be the Hessians as defined previously. The difference in the stable rank between $\boldsymbol{H}_N(h^R, \omega^R)$ and $\boldsymbol{H}_M(h^R, \omega^R)$ converges weakly almost surely to positive value $S(\theta_1, \cdots, \theta_{p+q}, \mu) > 0$ as $R \to \infty$, i.e.*

$$\text{srank}(\boldsymbol{H}_M(h^R, \omega^R)) - \text{srank}(\boldsymbol{H}_N(h^R, \omega^R)) \to S(\theta_1, \cdots, \theta_{p+q}, \mu) > 0. \tag{5}$$

*Here, the value $S(\theta_1, \cdots, \theta_{p+q}, \mu)$ does not dependent on the sequence $(h^R, \omega^R)$.*

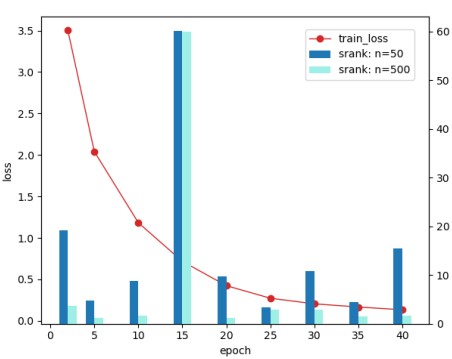
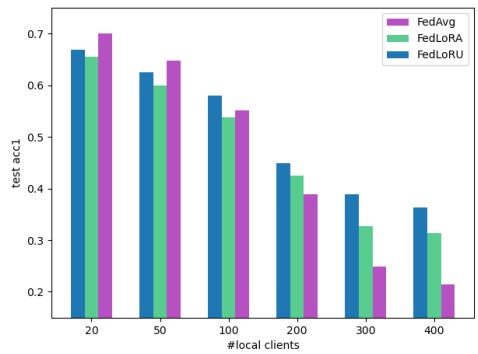

(a) Estimated stable rank for different dataset sizes          (b) Comparing test accuracy by number of clients

Figure 2: Figure 2(a) presents a comparison of the estimated stable rank of the Hessian for dataset sizes of 50 and 500. The estimated stable rank of the Hessian for the dataset size of 50 consistently exceeds that of the dataset size of 500. For an experiment detail, see Appendix C.3. Figure 2(b) illustrates the test accuracy of FedAvg and FedLoRU across varying numbers of clients.

Theorem 3.2 implies that individual clients in federated learning, working with smaller datasets, inherently have higher-rank structures in their local Hessians compared to the server's Hessian. This may lead to larger discrepancies across clients due to increased complexity and variability in local training landscape, causing more divergent optimization paths and complicating the aggregation process. Our empirical results in Figure 2 further support this by demonstrating that smaller datasets exhibit higher estimated stable ranks, and as the number of clients increases (i.e., local dataset size decreases), low-rank updates outperform full-rank updates.

Understanding this phenomenon is crucial for developing more effective federated learning algorithms. By acknowledging the higher rank structure of client's Hessian, constraining the rank of client-side optimization can mitigate the discrepancies, especially when local dataset sizes are very small. In the next section, we introduce an algorithm that leverages this insight.

## 3.2 FEDERATED LOW-RANK UPDATE (FEDLORU) ALGORITHM

Consider a federated learning system with $K$ clients, where each client $k$ has its own loss function $f^{(k)} : \mathbb{R}^{m \times n} \to \mathbb{R}$. The server aims to find a global model $\boldsymbol{W} \in \mathbb{R}^{m \times n}$ that minimizes the aggregated loss function $f(\boldsymbol{W}) = \sum_{k=1}^{K} p^{(k)} f^{(k)}(\boldsymbol{W})$, where $p^{(k)}$ is the weight of client $k$.

**Fedeated low-rank update algorithm**  To enhance communication efficiency, FedLoRU constraints clients' updates to low-rank. Analogous to the LoRA (Hu et al., 2021) approach, at each iteration, client $k$ holds a frozen local copy of the global model $\boldsymbol{W}$ and performs local training to find low-rank matrices $\boldsymbol{A}_t^{(k)} \in \mathbb{R}^{m \times r}$ and $\boldsymbol{B}_t^{(k)} \in \mathbb{R}^{r \times n}$ by solving:

$$\boldsymbol{A}_t^{(k)}, \ \boldsymbol{B}_t^{(k)} = \arg\min_{\boldsymbol{A}, \ \boldsymbol{B}} f^{(k)}(\boldsymbol{W} + \alpha \boldsymbol{A}\boldsymbol{B}) \tag{6}$$

where $\alpha$ is a fixed scaling hyperparameter. At the end of each iteration, the server collects $\boldsymbol{A}_t^{(k)}$ and $\boldsymbol{B}_t^{(k)}$ and aggregates them by averaging: $\boldsymbol{A}_t = \sum_{k \in \mathcal{K}_M} p^{(k)} \boldsymbol{A}_t^{(k)}$, $\boldsymbol{B}_t = \sum_{k \in \mathcal{K}_M} p^{(k)} \boldsymbol{B}_t^{(k)}$. After the aggregation, the server broadcasts $\boldsymbol{A}_t$ and $\boldsymbol{B}_t$ to the clients, who continue local training using these matrices.

However, unlike LoRA, FedLoRU accumulates low-rank updates into the global model after aggregation to achieve a higher-rank global model. Clients subsequently update their local copies of the global model by $\boldsymbol{W} \leftarrow \boldsymbol{W} + \alpha \boldsymbol{A}_t \boldsymbol{B}_t$ and reset their low-rank matrices. When low-rank updates are accumulated every $\tau$ rounds from the initial global model $\boldsymbol{W}$, the final global model at round $T$ is $\boldsymbol{W}_T = \boldsymbol{W} + \sum_{\substack{t=1 \\ t \bmod \tau = 0}}^{T} \boldsymbol{A}_t \boldsymbol{B}_t$.

---

**Algorithm 1** FedLoRU. $\boldsymbol{W}$ is a model, $\boldsymbol{A}_0, \boldsymbol{B}_0$ are initial low-rank update matrices, $\alpha$ is a scaling factor, $\tau$ is an accumulation cycle, $T$ is the total training round.

---

**Require:** $\boldsymbol{W}, \boldsymbol{A}_0, \boldsymbol{B}_0, \alpha, \tau, T$.
  **Initialize:** Server sends $\boldsymbol{W}$ to each client.
  **for** $t = 1, \cdots, T$ **do**
    Server selects $M$ clients $\mathcal{K}_M$ and distributes $\boldsymbol{A}_{t-1}, \boldsymbol{B}_{t-1}$ to clients in $\mathcal{K}_M$.
    **for** each client $k \in \mathcal{K}_M$ **do**
      **Local training:** Find $\boldsymbol{A}_t^{(k)}, \boldsymbol{B}_t^{(k)}$ by solving (6) starting from $\boldsymbol{A}_{t-1}, \boldsymbol{B}_{t-1}$.
      Send $\boldsymbol{A}_t^{(k)}, \boldsymbol{B}_t^{(k)}$ to the server.
    **end for**
    **Server aggregation:** $\boldsymbol{A}_t \leftarrow \sum_{k \in \mathcal{K}_M} p^{(k)} \boldsymbol{A}_t^{(k)}, \boldsymbol{B}_t \leftarrow \sum_{k \in \mathcal{K}_M} p^{(k)} \boldsymbol{B}_t^{(k)}$.
    **if** $t \bmod \tau = 0$ **then**
      Server distributes $\boldsymbol{A}_t, \boldsymbol{B}_t$ to all clients .
      Each client $k$ updates its local copy of the global model: $\boldsymbol{W} \leftarrow \boldsymbol{W} + \alpha \boldsymbol{A}_t \boldsymbol{B}_t$.
    **end if**
  **end for**
  **Return:** $\boldsymbol{W} + \alpha \sum_{t=1: \, t \bmod \tau = 0}^{T} \boldsymbol{A}_t \boldsymbol{B}_t$.

---

FedLoRU enables training a higher-rank global model alongside low-rank local updates. With each accumulation of low-rank update matrices, the global model's rank is incrementally enhanced, enabling the initiation of new learning phases. Moreover, constraining the rank of local training introduces a regularization effect, thereby diminishing the discrepancy between updated local models.

**Communication overhead** FedLoRU reduces communication overhead from $Kmn$ to $Kr(m + n)$ when $r \ll m$ or $r \ll n$. While we use a low-rank factorized model, alternatives like LoKr or LoHa can be employed, differing only in the factorization scheme but based on the same principles. Additionally, since no compression process is involved, there is no additional computation compared to conventional compression-based communication-efficient federated learning algorithms.

**Federated low-rank update for statistical and model heterogeneous setting** We develop the personalized FedLoRU (pFedLoRU) algorithm to address statistical heterogeneity (non-iid) in federated learning, building on the FedLoRU approach. In pFedLoRU, each client $k$ maintains a local copy of the global model $\boldsymbol{W}$, global low-rank matrices $\boldsymbol{A}^{(k)}$ and $\boldsymbol{B}^{(k)}$, and personal matrices $\boldsymbol{L}^{(k)}$ and $\boldsymbol{U}^{(k)}$. The matrices $\boldsymbol{A}^{(k)}$ and $\boldsymbol{B}^{(k)}$ are shared with the server to update the global model, while $\boldsymbol{L}^{(k)}$ and $\boldsymbol{U}^{(k)}$ are tailored to adapt to the local distribution. In each round $t$, client $k$ optimizes the personal matrices for $E_{\text{per}}$ epochs and the global matrices for $E_{\text{global}}$ by solving:

$$\boldsymbol{L}_t^{(k)}, \ \boldsymbol{U}_t^{(k)} = \underset{\boldsymbol{L}, \, \boldsymbol{U}}{\arg\min} \, f^{(k)}(\boldsymbol{W} + \alpha_{\text{global}} \boldsymbol{A}_{t-1} \boldsymbol{B}_{t-1} + \alpha_{\text{per}} \boldsymbol{L} \boldsymbol{U}) \tag{7}$$

$$\boldsymbol{A}_t^{(k)}, \ \boldsymbol{B}_t^{(k)} = \underset{\bar{\boldsymbol{A}}, \, \bar{\boldsymbol{B}}}{\arg\min} \, f^{(k)}(\boldsymbol{W} + \alpha_{\text{global}} \bar{\boldsymbol{A}} \bar{\boldsymbol{B}} + \alpha_{\text{per}} \boldsymbol{L}_t^{(k)} \boldsymbol{U}_t^{(k)}) \tag{8}$$

Subsequently, the server collects the global update matrices $\boldsymbol{A}_t^{(k)}$ and $\boldsymbol{B}_t^{(k)}$ from the clients, performs aggregation $\boldsymbol{A}_t \leftarrow \sum_{k \in \mathcal{K}_M} p^{(k)} \boldsymbol{A}_t^{(k)}, \boldsymbol{B}_t \leftarrow \sum_{k \in \mathcal{K}_M} p^{(k)} \boldsymbol{B}_t^{(k)}$, and broadcasts $\boldsymbol{A}_t$ and $\boldsymbol{B}_t$ to the clients. The clients then accumulate the low-rank updates accordingly as in FedLoRU.

On the other hand, when local clients possess varying hardware resources, it becomes impractical to use uniform low-rank matrices across all clients. To address this issue, we develop the model-heterogeneous FedLoRU (mFedLoRU) algorithm, which employs hierarchical low-rank updates that allows clients to use their adaptive update ranks.

In mFedLoRU, at each round $t$, each client $k$ receives $\boldsymbol{A}_{t-1}$ and $\boldsymbol{B}_{t-1}$ and updates its local copy of the global model as in FedLoRU. For local training, each client $k$ generates and optimizes the nested low-rank matrices $\boldsymbol{A}_{\text{d}}^{(k)} \boldsymbol{A}_{\text{u}}^{(k)}$ and $\boldsymbol{B}_{\text{d}}^{(k)} \boldsymbol{B}_{\text{u}}^{(k)}$ by solving:

$$\boldsymbol{A}_{\text{d}}^{(k)}, \boldsymbol{A}_{\text{u}}^{(k)}, \boldsymbol{B}_{\text{d}}^{(k)}, \boldsymbol{B}_{\text{u}}^{(k)} = \underset{\bar{\boldsymbol{A}}_{\text{d}}, \bar{\boldsymbol{A}}_{\text{u}}, \bar{\boldsymbol{B}}_{\text{d}}, \bar{\boldsymbol{B}}_{\text{u}}}{\arg\min} \, f^{(k)}(\boldsymbol{W} + \alpha(\boldsymbol{A}_{t-1} + \alpha_A^{(k)} \bar{\boldsymbol{A}}_{\text{d}} \bar{\boldsymbol{A}}_{\text{u}})(\boldsymbol{B}_{t-1} + \alpha_B^{(k)} \bar{\boldsymbol{B}}_{\text{d}} \bar{\boldsymbol{B}}_{\text{u}})). \tag{9}$$

Table 1: Top-1 test accuracy comparison with different communication-efficient federated learning methods under various FL settings. The parameter ratio refers to the proportion of trainable parameters in the model compared to the full-rank model used in FedAvg.

(a) Fashion-MNIST

| Setting | IID - #clients=20 | | | IID - #clients=100 | | | NonIID - #clients=20 | | |
|---|---|---|---|---|---|---|---|---|---|
| **Param Ratio** | **44%** | **33%** | **22%** | **44%** | **33%** | **22%** | **44%** | **33%** | **22%** |
| FedLoRA | 91.22 | 90.29 | 90.15 | 88.63 | 88.14 | 88.01 | 73.89 | 74.00 | 73.19 |
| FedHM | 91.16 | 91.10 | **90.94** | **89.43** | **89.37** | **88.86** | 85.15 | **85.45** | **85.33** |
| FedLoRU | **91.25** | **91.16** | 90.59 | 89.01 | 88.88 | 88.37 | **85.33** | 80.02 | 80.17 |

(b) CIFAR-10

| Setting | IID - #clients=20 | | | IID - #clients=100 | | | NonIID - #clients=20 | | |
|---|---|---|---|---|---|---|---|---|---|
| **Param Ratio** | **41%** | **31%** | **21%** | **41%** | **31%** | **21%** | **41%** | **31%** | **21%** |
| FedLoRA | 91.65 | 88.96 | 89.35 | 79.48 | 85.71 | 85.06 | 69.60 | 66.13 | 67.61 |
| FedHM | 90.76 | 90.32 | 90.77 | 81.41 | 81.58 | 82.12 | 70.55 | 66.39 | 65.48 |
| FedLoRU | **92.43** | **90.71** | **90.85** | **81.46** | **86.01** | **86.10** | **75.19** | **69.71** | **67.88** |

(c) CIFAR-100

| Setting | IID - #clients=20 | | | IID - #clients=100 | | | NonIID - #clients=20 | | |
|---|---|---|---|---|---|---|---|---|---|
| **Param Ratio** | **41%** | **31%** | **21%** | **41%** | **31%** | **21%** | **41%** | **31%** | **21%** |
| FedLoRA | 65.53 | 57.36 | 55.14 | 53.79 | 52.20 | 51.20 | 14.41 | 10.58 | 12.97 |
| FedHM | 59.43 | 58.40 | 58.52 | 43.35 | 41.84 | 41.62 | **16.88** | 15.04 | 14.13 |
| FedLoRU | **66.81** | **60.78** | **61.42** | **57.96** | **53.25** | **53.53** | 16.46 | **15.70** | **14.52** |

Here, $A_{t-1}B_{t-1}$ are the rank-$r$ low-rank matrices, and $A_{\mathrm{d}}^{(k)}A_{\mathrm{u}}^{(k)}$ and $B_{\mathrm{d}}^{(k)}B_{\mathrm{u}}^{(k)}$ are rank-$r_A$ and rank-$r_B$ low-rank matrices used to update $A_{t-1}$ and $B_{t-1}$. After local training, the server collects $A_d^{(k)}, A_u^{(k)}$, recovers the low-rank update matrix $A_t^{(k)} \leftarrow A_{t-1} + \alpha_A^{(k)} A_d^{(k)} A_u^{(k)}$, and finally aggregates $A_t \leftarrow \sum_{k \in \mathcal{K}_M} p^{(k)} A_{t-1}^{(k)}$. The same process applies for the low-rank matrices $B_d^{(k)}$ and $B_d^{(k)}$. A detailed description of pFedLoRU and mFedLoRU algorithm can be found in Appendix B.

## 4 EXPERIMENTS

In this section, we extensively evaluate FedLoRU on pre-training and fine-tuning on different homogeneous and heterogeneous settings. We first provide the experiment setup such as baselines and heterogeneous settings, then move on to the performance evaluation.

### 4.1 EXPERIMENT SETUP

**Datasets and Baseline Algorithms** We evaluate our proposed algorithms on four standard datasets: Fashion MNIST (Xiao et al., 2017), CIFAR-10, CIFAR-100 (Krizhevsky et al., 2009), and Alpaca (Taori et al., 2023). ResNet-10 and ResNet-18 (He et al., 2016) are used for the image datasets, and LLaMA2-3B (Touvron et al., 2023) is used for the language dataset. We compare FedLoRU with several benchmarks: FedAvg (McMahan et al., 2017), the standard federated learning algorithm that trains full-rank models; FedLoRA (Zhang et al., 2023), which trains low-rank modules without accumulating low-rank updates; and FedHM (Yao et al., 2021), the prior state-of-the-art in communication-efficient federated learning. For pFedLoRU, we compare against pFedLoRA (Wu et al., 2024), and for mFedLoRU, we compare with the model-heterogeneous version of FedHM.

**Implementation** During pre-training on the image datasets, we vary the number of clients between 20 and 400, sampling 50% of clients per round, as is standard in FL literature, with each client training for 5 local epochs. For fine-tuning the language model, we use 10 clients with a 50% participation and 1 local epoch. The selection of local epochs balances the trade-off between communication overhead and potential performance degradation. Learning rates are selected via grid search, and different rank configurations are tested for FedHM, FedLoRA, and FedLoRU. In

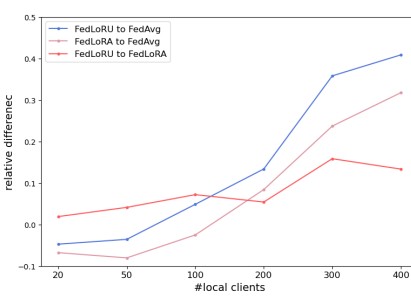

(a) Relative performance by $K$

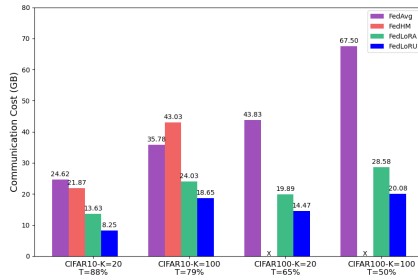

(b) Communication cost for target accuracy

Figure 3: Figure 3(a) presents the relative difference in test accuracy between two algorithms in terms of the number of clients $K$. For example, the ralative difference of FedLoRU to FedAvg is defined as $\frac{\text{FedLoRU}-\text{FedAvg}}{\text{FedLoRU}}$. For the detailed number, see Appendix D.4. Figure 3(b) evaluates different low-rank federated learning algorithms in terms of the communication cost to achieve target test accuracy. Here, "X" indicates that the algorithm did not reach the target accuracy.

fact, while we use FedAvg as the training scheme, FedLoRU techniques can be easily integrated into other federated learning schemes such as FedAdam and FedAdagrad (Reddi et al., 2020). For full details of the implementation, including choice of $\alpha$, $\tau$, $T$, see Appendix C.

In the statistically heterogeneous setting, we generate disjoint Non-IID client data using a Dirichlet distribution, $\text{Dir}(\psi)$, with a concentration parameter $\psi$ set to 0.5, as described in Hsu et al. (2019). For the model heterogeneous setting, we simulate virtual environments where each client is assigned a different nominal rank, thereby restricting them to use low-rank update matrices of varying ranks. The specific configurations for these settings are detailed in Table A3.

## 4.2 PERFORMANCE EVALUATION

**Performance of Pre-training**   We evaluate the Top-1 accuracy of models with varying parameter sizes in both IID and Non-IID scenarios across different federated learning configurations. Table 1 shows the performance of FedLoRU and baseline algorithms.

In our experimental evaluation, FedLoRU consistently achieves competitive or superior accuracy compared to FedAvg, whose results can be found in Appendix D. Although FedLoRU's accuracy is slightly lower than FedAvg's in most settings, the difference is minimal given the significant reduction in parameters, with at most a 5% decrease and typically only a 1-2% difference. Notably, in the CIFAR-10 and CIFAR-100 IID settings with 100 clients, FedLoRU surpasses FedAvg. Overall, FedLoRU achieves the best accuracy in 20 out of 27 cases and demonstrates improvements over FedHM ranging from -6% to 33.7%. Additionally, FedLoRU consistently outperforms FedLoRA, highlighting the effectiveness of accumulating low-rank updates. The client regularization effect of FedLoRU, as predicted by our theoretical analysis, suggests that using client-side low-rank updates is particularly beneficial in environments with a large number of clients. This benefit is evident in experiments under IID conditions with 100 clients, where FedLoRU attains the highest accuracy among the tested methods.

**Scalability and Performance of FedLoRU in Large-Client Federated Learning**   Table A5 and Figure 3(a) compare FedAvg and FedLoRU in varying cross-device FL settings, where many small edge devices collaboratively train a model. As the number of clients increases, the scalability of algorithms become critical. Our experiments show a sharp decline in FedAvg's performance, with Top-1 accuracy dropping from 69.97% at $K = 20$ to just 21.44% at $K = 400$. This indicates FedAvg struggles to maintain accuracy in cross-device FL.

In contrast, FedLoRU outperforms FedAvg as the number of clients increases. While FedAvg slightly outperforms FedLoRU with smaller numbers of clients ($K = 20$ and $K = 50$), FedLoRU consistently surpasses FedAvg when the client count ranges from $K = 100$ to $K = 400$. The widening performance gap as $K$ increases highlights FedLoRU's superior scalability and effectiveness in large-scale federated learning, especially with extensive client participation.

**Performance of LLM Fine-tuning** Figure 4 presents the loss curves of FedLoRA and FedLoRU during the fine-tuning of a LLaMA2-3B model on the Alpaca dataset. The train loss curves show that both algorithms achieve similar convergence rates, with minimal differences in training optimization. However, a notable distinction emerges in the test loss results, where FedLoRU consistently outperforms FedLoRA after the 25th communication round.

In this fine-tuning experiment, we accumulate the results every 15 communication rounds. Notably, despite FedLoRU performing an additional accumulation at round 30, the test loss does not show any further improvement. This suggests that beyond a certain point, further accumulation may not necessarily enhance the model's generalization performance.

**Performance of pFedLoRU and mFedLoRU** In our experiments, we evaluate the performance of pFedLoRU and mFedLoRU on statistical heterogeneous and model heterogeneous FL environments. Table 2 shows the performance of pFedLoRU and pFedLoRA. Under both non-IID levels ($\psi = 0.1$ and $\psi = 0.5$), pFedLoRU shows a clear advantage in terms of accuracy compared to pFedLoRA. In addition, despite having less than half the number of parameters, pFedLoRU consistently achieves higher accuracy.

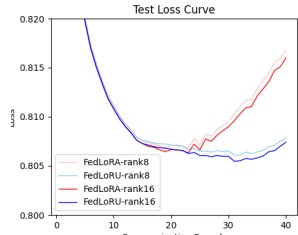

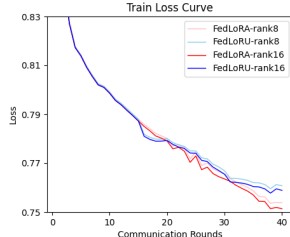

Figure 4: Loss curve of FedLoRU and FedLoRA for fine-tuning LLaMA2-3B.

Table 2: Comparison of pFedLoRA and pFedLoRU with varying non-iidness ($\psi$) on CIFAR100.

| Algorithm | #params | Non-IIDness | |
|---|---|---|---|
| | | $\psi = 0.1$ | $\psi = 0.5$ |
| **pFedLoRA(1)** | 11.22M | 45.36 | 42.14 |
| **pFedLoRA(2)** | 11.22M | 47.45 | 42.28 |
| **pFedLoRU** | 4.63M | **49.65** | **46.50** |

Table 3: Comparison of FedHM and mFed-LoRU in two model-heterogeneous setting.

| Dataset | Setting | FedHM | mFedLoRU |
|---|---|---|---|
| **CIFAR-10** | setting 1 | **88.09** | 84.81 |
| | setting 2 | **88.68** | 84.36 |
| **CIFAR-100** | setting 1 | 49.84 | **51.16** |
| | setting 2 | 50.52 | **50.89** |

On the other hands, Table 3 shows the performanec of mFedLoRU and FedHM. FedHM outperforms mFedLoRU in both heterogeneous settings (setting 1 and setting 2) for the CIFAR-10 dataset, indicating that FedHM handles model heterogeneity more effectively for simpler tasks. This suggests that FedHM is better suited for less complex datasets such as CIFAR-10, where its approach proves more efficient. However, mFedLoRU outperforms FedHM in both heterogeneous settings for the more complex CIFAR-100 dataset, demonstrating its potential in addressing the model-heterogeneous problem in federated learning. A key advantage of mFedLoRU is that it does not require additional computational steps, such as the weight factorization used in FedHM, making it a more efficient solution in scenarios involving more challenging tasks.

## 5 CONCLUSION

In this paper, we theoretically show that client-side optimization exhibits a higher-rank structure compared to server-side optimization and hypothesize that using low-rank updates in client-side optimization can promote an implicit regularization effect across clients. We are the first to establish a theoretical foundation supporting the use of low-rank updates in federated learning. Our proposed algorithm, FedLoRU, achieves comparable performance to FedAvg while significantly reducing the number of communicated parameters. Moreover, as the number of clients increases, FedLoRU consistently outperforms FedAvg, highlighting its scalability and effectiveness in large-scale federated learning environments.

We further extend our approach by introducing two algorithm variants: pFedLoRU and mFedLoRU, which generalize low-rank updates to address statistical and model heterogeneity, respectively. Future work can focus on investigating the relationship between the accumulation schedule and performance in FedLoRU, as well as exploring the connection between different ranks and accumulation schedules to further optimize the algorithm's efficiency and performance.

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

# A    PROOF OF THE MAIN THEOREM

In this section, we provide proofs of Proposition A.4 Proposition 3.1 and Theorem 3.2. We begin by presenting some lemmas that will be required for our analysis, then proceed to prove the propositions and the theorem.

**Lemma A.1** (Theorem 2.2 from Pielaszkiewicz & Singull (2015)). *Let $\mu_n$ be a sequence of probability measures on $\mathbb{R}$ and let $g_{\mu_n}$ denote the Stieltjes transform of $\mu_n$. Then*

*a) if $\mu_n \to \mu$ weakly, where $\mu$ is a measure on $\mathbb{R}$, then $g_{\mu_n}(z) \to g_\mu(z)$ pointwise for any $z \in \{z : z \in \mathbb{C}, \ \mathfrak{I}(z) > 0\}$*

*b) if $g_{\mu_n}(z) \to g(z)$ pointwise, for all $z \in \{z : z \in \mathbb{C}, \ \mathfrak{I}(z) > 0\}$, then there exists a unique non-negative and finite measure such that $g = g_\mu$ and $\mu_n \to \mu$ weakly*

**Lemma A.2** (Theorem 3.4 from Baskerville et al. (2022)). *Let $X$ be an $N \times N$ real symmetric random matrix and let $D$ be an $N \times N$ symmetric matrix(deterministic or random). Let $\widehat{\mu}_X, \widehat{\mu}_D$ be the empirical spectral measures of the sequence of matrices $X, D$ and assume there exist deterministic limit measures $\mu_X, \mu_D$. Assume that $X$ has QUE and $\widehat{\mu}_X$ concentrates in the sense that*

$$\mathbb{P}(W_1(\widehat{\mu}_X, \mu_X) > \delta) \leq e^{-N^\tau f(\delta)}$$

*where $\tau > 0$ and $f$ is some positive increasing function. Then $H = X + D$ has a limiting spectral measure and it is given by the free convolution $\mu_X \mu_D$*

**Lemma A.3** (Weyl's inequality). *For Hermitian matrices $\boldsymbol{A}, \boldsymbol{B} \in \mathbb{C}^{R \times n}$ and $i, j \in \{1, c \ldots, n\}$,*

$$\lambda_{i+j-1}(\boldsymbol{A} + \boldsymbol{B}) \leq \lambda_i(\boldsymbol{A}) + \lambda_j(\boldsymbol{B}), \quad i + j \leq n + 1, \tag{10}$$

$$\lambda_{i+j-n}(\boldsymbol{A} + \boldsymbol{B}) \geq \lambda_i(\boldsymbol{A}) + \lambda_j(\boldsymbol{B}), \quad i + j \geq n + 1, \tag{11}$$

*where $\lambda_i(\boldsymbol{A})$ is $i$-th eigenvalue of $\boldsymbol{A}$.*

A.1 FURTHER DISCUSSION ON $\Omega(\theta_1, \cdots, \theta_k)$.

In our theoretical analysis, we show the difference in stable rank between the Hessians of a server and a client eventually becomes positive as dimension $R$ approaches infinity within the space of $\Omega(\theta_1, \cdots, \theta_k)$. In this section, we will discuss about the richness of $\Omega(\theta_1, \cdots, \theta_k)$ and characteristics of $\Omega^R(\theta_1, \cdots, \theta_k)$. They are defined as:

$$\Omega^R(\theta_1, \cdots, \theta_k) = \{(h^R, \omega^R) : \boldsymbol{H}_{\text{true}}(h^R, \omega^R) \text{ has eigenvalues } \theta_1, \cdots, \theta_k\}, \tag{12}$$

$$\Omega(\theta_1, \cdots, \theta_k) = \bigcup_R \Omega^R(\theta_1, \cdots, \theta_k). \tag{13}$$

In fact, the set of all possible pairs $(h^R, \omega^R)$ is represented by the union over all dimensions $R$, integers $k \leq R$, and non-zero real values $\theta_1, \cdots, \theta_k$ as follows:

$$\{(h^R, \omega^R) : \text{dimension } R < \infty\} = \bigcup_{R=1}^{\infty} \bigcup_{k=1}^{R} \bigcup_{(\theta_1, \cdots, \theta_k) \in \mathbb{R}^k} \Omega^R(\theta_1, \cdots, \theta_k).$$

Thus, for any given pair $(h^R, \omega^R)$, there exist $\theta_1, \cdots, \theta_k$ such that $(h^R, \omega^R) \in \Omega^R(\theta_1, \cdots, \theta_k)$. According to the following proposition, either the set $\Omega(\theta_1, \cdots, \theta_k)$ is empty or there exist infinitely many values of $R$ for which $\Omega^R(\theta_1, \cdots, \theta_k) \neq \emptyset$.

**Proposition A.4.** *Let $\theta_1, \cdots, \theta_k$ be fixed non-zero real numbers, and suppose there exists $\tilde{R} > k$ such that $\Omega^{\tilde{R}}(\theta_1, \cdots, \theta_k)$ is non-empty. Then there are infinitely many $R$ such that $\Omega^R(\theta_1, \cdots, \theta_k)$ is non-empty. In particular, $\Omega^R(\theta_1, \cdots, \theta_k)$ is non-empty for all $R \geq \tilde{R}$.*

*Proof.* Suppose $(h^{\tilde{R}}, \omega^{\tilde{R}}) \in \Omega^{\tilde{R}}(\theta_1, \cdots, \theta_k)$, i.e., $\boldsymbol{H}_{\text{true}}(h^{\tilde{R}}, \omega^{\tilde{R}})$ has non-zero eigenvalues $\theta_1, \cdots, \theta_k$. To construct a prediction function $h^{\tilde{R}+1}$ and a weight $\omega^{\tilde{R}+1}$ of dimension $\tilde{R} + 1$ such that the true Hessian retains the same non-zero eigenvalues, define:

$$h^{\tilde{R}+1}(\omega^{\tilde{R}}, z) = h^{\tilde{R}}(\omega^{\tilde{R}}) + g^{\tilde{R}+1}(\omega^{\tilde{R}}, z) \tag{14}$$

$$\omega^{\tilde{R}+1} = (\omega^{\tilde{R}}, z) \tag{15}$$

where $\nabla^2 \int \ell(g^{\tilde{R}+1}(\boldsymbol{x}; (\omega^{\tilde{R}}, z)), \boldsymbol{y}) d\psi(\boldsymbol{x}, \boldsymbol{y}) = 0$ ensures that the second derivative with respect to the new function $g^{\tilde{R}+1}$ vanishes. Thus, since $h^{\tilde{R}+1}$ and $h^{\tilde{R}}$ share the same true Hessian, except for the intersecting zero-row and zero-column—which have no impact on the eigenvalues of the Hessian—it follows that $(h^{\tilde{R}+1}, \omega^{\tilde{R}+1}) \in \Omega^{\tilde{R}+1}(\theta_1, \cdots, \theta_k)$. Specifically, if we consider feed-forward neural networks as prediction functions, one can easily construct a larger neural network that maintains the same non-zero eigenvalues by adding an additional neuron with a single connection to a neuron in the previous layer. This additional neuron does not affect the final output, thereby preserving the desired eigenvalue properties. $\square$

## A.2 PROOF OF PROPOSITION 3.1

To prove Proposition 3.1, we decompose the eigenvalue analysis into two distinct parts. First, we demonstrate that the $i$-th eigenvalues, where $i \in \{p+1, \cdots, P-q-1\}$, converge to the upper or lower bounds of the spectral density of $\mu_N$. This portion of the proof parallels the approach employed by Benaych-Georges & Nadakuditi (2011). Second, we show that the remaining eigenvalues converge to the Stieltjes transformation. This part of the proof follows the methodology outlined by Baskerville et al. (2022).

*Proof.* In the proof, we drop dependency on $(h^R, \omega^R)$ since it is clear. First, consider $\lambda_i(\boldsymbol{H}_N)$ where $p < i < R - q$. By using Lemma A.3, we have

$$\lambda_i(\boldsymbol{H}_N) \leq \lambda_{1+i-j}(\boldsymbol{H}_{\text{true}}) + \lambda_{1+i-k}(\epsilon(N)), \quad i \leq R,\ i = j + k - 1,\ j, k \in \{1, \cdots, R\} \quad (16)$$

$$\lambda_i(\boldsymbol{H}_N) \leq \lambda_{R+i-j}(\boldsymbol{H}_{\text{true}}) + \lambda_{R+i-k}(\epsilon(N)), \quad i \geq 1,\ i = j + k - R,\ j, k \in \{1, \cdots, R\} \quad (17)$$

If we put $k = 1 + p$ on (16) and $k = R - q$ on (17), since $\lambda_{1+p}(\boldsymbol{H}_{\text{true}}) = 0$ and $\lambda_{R-q}(\boldsymbol{H}_{\text{true}}) = 0$, we deduce

$$\lambda_{i+q}(\epsilon(N)) \leq \lambda_i(H_N) \leq \lambda_{i-p}(\epsilon(N)), \quad \forall i \in \{1, \cdots, R\} \quad (18)$$

where $\lambda_k(\epsilon(N)) = -\infty$ if $k > R$ and $+\infty$ if $k \leq 0$. Additionally, since $\epsilon(N)$ has the limiting spectral density $\mu_N$ and $L_N, U_N$ are lower and upper bound of $\mu_N$, we have for all $1 \leq i \leq R$,

$$\liminf_{R \to \infty} \lambda_i(\epsilon(N)) \geq U_N \quad \text{and} \quad \limsup_{R \to \infty} \lambda_{R+1-i}(\epsilon(N)) \leq L_N \quad (19)$$

$$\lambda_1(\epsilon(N)) \to U_N \quad \text{and} \quad \lambda_P(\epsilon(N)) \to L_N \quad (20)$$

From the above relations, it follows that for all fixed $1 \leq i \leq R$, $\lambda_i(\epsilon(N)) \to U_N$ and $\lambda_{R+1-i}(\epsilon(N)) \to L_N$. By (18), we have

$$\liminf_{n \to \infty} \lambda_i(H_N) \geq U_N \quad \text{and} \quad \limsup_{n \to \infty} \lambda_i(H_N) \leq L_N \quad (21)$$

and for all $i > p$ (resp. $i \geq q$) fixed, we have

$$\lambda_i(H_N) \to U_N \quad (\text{resp. } \lambda_{R-i}(H_N) \to L_N) \quad (22)$$

Now, we are going to prove the remaining eigenvalues $\lambda_i(\boldsymbol{H}_N)$, where $i \in \{1, \cdots, p, R - q + 1, \cdots R\}$. Note that, since $p + q \ll R$ when $R$ is large enough, the limiting spectral density of $H_{true}$ converges to $\nu = \delta_0$.

Consider $\lambda_i(H_N)$ where $i \leq p$ or $i \geq R - q$. By the Lemma A.2, the limiting spectral density $\mu_{H_N}$ of $H_N$ is $\mu_N \nu$ where $\mu_N$ is the limiting spectral density of $\epsilon(N)$. Then by the Lemma A.1, the Stieltjes transform $g_{\mu_{H_N}}(z)$ converges pointwise to $g_{\nu \mu_N}(z)$ for any $z \in \{z : z \in \mathbb{C}, \Im(z) > 0\}$. Therefore, we have

$$\begin{aligned}
\widehat{g}_{H_N(h^R, \omega^R)}(z) &= g_{\mu_{H_N}(h^R, \omega^R)}(z) + o(1) \\
&= g_{\mu_N(h^R, \omega^R)\nu(h^R, \omega^R)}(z) + o(1) \\
&= g_{\nu(h^R, \omega^R)}(k(z)) + o(1) \\
&= \widehat{g}_{H_{\text{true}}(h^R, \omega^R)}(k(z)) + o(1)
\end{aligned} \quad (23)$$

where $k$ is the subordination function such that $g_{\mu_N \nu}(z) = g_\nu(k(z))$.

Let $\lambda \in \mathbb{R} \backslash \mathrm{supp}(\mu_N \nu)$ be an eigenvalue of $H_N$. Then $\widehat{g}_{H_N}$ has a singularity at $\lambda$, thus $\widehat{g}_{H_{\mathrm{true}}}$ has a singularity at $k(\lambda)$, thus, for any $R$, this singularity should persist and $k(\lambda)$ must coincide with one of the outliers of $H_N$, i.e., $\theta_i$ is an outlier eigenvalue of $H_{true}$ if and only if there exists an eigenvalue $\lambda$ of $H_N$ contained in $\mathbb{R} \backslash \mathrm{supp}(\mu_N \nu)$ such that $k(\lambda) = \theta_i$. Thus, we can write the outliers of $H_N$ as

$$\{k^{-1}(\theta_j) : k^{-1}(\theta_j) \in \mathbb{R} \backslash \mathrm{supp}(\mu_N \nu)\} \tag{24}$$

Note that $\mathrm{supp}(\mu_N \nu) = \mathrm{supp}(\mu_N \delta_0) = \mathrm{supp}(\mu_N)$. Now, we want to find the form of $k^{-1}(\theta_j)$. From the subordination function relation, we have

$$\begin{aligned} k^{-1}(\theta) &= g_{\mu_N \nu}^{-1}(g_\nu(\theta)) \\ &= \mathcal{R}_{\mu_N}(g_\nu(\theta) + g_\nu^{-1}(g_\nu(\theta))) \\ &= \mathcal{R}_{\mu_N}(1/\theta) + \theta \end{aligned} \tag{25}$$

Note that by the definition of Stieltjes transformation and $\mathcal{R}$-transform $g_\nu(\theta) = g_{\delta_0}(\theta) = 1/\theta$.

Let $m_n^{(\mu)}$ be the $n$-th moment of a distribution $\mu$ and $C_n^{(\mu)}$ be the $n$-th cumulant of $\mu$. Then we have the relationship between $m_n^{(\mu)}$ and $C_n^{(\mu)}$ ([3]) as

$$m_n^{(\mu)} = \sum_{r=1}^{n} \sum_{\substack{0 \le i_1, \cdots, i_r \le n-r \\ i_1 + \cdots + i_r = n-r}} C_r^{(\mu)} \left[ \Pi_{j=1}^r m_{i_j}^{(\mu)} \right] \tag{26}$$

Therefore, from the moment's scaling property, $m_n^{\mu_N} = s(N)^n m_n^\mu$, we can deduce the scaling relation property of the cumulants, $C_n^{(\mu_N)} = s(N)^n C_n^{(\mu)}$, therefore we have the scaling property of $\mathcal{R}$-transform:

$$\mathcal{R}_{\mu_N}(\theta) = s(N) \mathcal{R}_\mu(s(N)\theta) \tag{27}$$

Finally, we have a expression for the outliers of $H_N$ as

$$k^{-1}(\theta) = s(N) \mathcal{R}_\mu(s(N)/\theta) + \theta \tag{28}$$

$\square$

### A.3 PROOF OF THEOREM 3.2

*Proof.* Suppose $\alpha$ and $\beta$ be the size of sets $\{i > p : \lambda_i(H_N) \to U_N\}$ and $\{i \ge q : \lambda_{R-i}(H_N) \to L_N\}$ respectively, and $a_N$ and $b_N$ be integers such that

$$\begin{aligned} g_N^{-1}(\theta_{a_N}) &> U_N > g_N^{-1}(\theta_{a_N+1}) \\ g_N^{-1}(\theta_{p+q-b_N}) &> L_N > g_N^{-1}(\theta_{p+q-b_N+1}) \end{aligned}$$

and we can define $a_M$ and $b_M$ in the similar manner. Then we have

$$\overbrace{\theta_1 > \cdots > \theta_{a_N}}^{a_N} > \overbrace{\theta_{a_N+1} > \cdots > \theta_p}^{p-a_N} > 0 > \overbrace{\theta_{p+1} > \cdots > \theta_{p+q-b_N}}^{q-b_N} > \overbrace{\theta_{p+q-b_N+1} > \cdots > \theta_{p+q}}^{b_N} \tag{29}$$

$$\overbrace{\theta_1 > \cdots > \theta_{a_M}}^{a_M} > \overbrace{\theta_{a_M+1} > \cdots > \theta_p}^{p-a_M} > 0 > \overbrace{\theta_{p+1} > \cdots > \theta_{p+q-b_M}}^{q-b_M} > \overbrace{\theta_{p+q-b_M+1} > \cdots > \theta_{p+q}}^{b_M} \tag{30}$$

WLOG, we can assume $\|\theta_1\| > \|\theta_{p+q}\|$ and define $g_M^{-1}(\theta_j) = \theta_j + s(M)\mathcal{R}_\mu(s(M)\theta_j^{-1})$. We will consider the limiting stable rank of the Hessians. From Proposition 3.1, the stable ranks of Hessians converges to the limiting stable ranks $\hat{\mathrm{srank}}(H_N)$ and $\hat{\mathrm{srank}}(H_M)$. Since $a_N > a_M$ and $b_N < b_M$, we can express the difference of the limiting stable rank of $H_N$ and $H_M$ as

$$\hat{\mathrm{srank}}(H_M) - \hat{\mathrm{srank}}(H_N)$$
$$= \sum_{j=2}^{a_M} \left\{ \left( \frac{g_M^{-1}(\theta_j)}{g_M^{-1}(\theta_1)} \right)^2 - \left( \frac{g_N^{-1}(\theta_j)}{g_N^{-1}(\theta_1)} \right)^2 \right\} + \sum_{j=a_M+1}^{a_N} \left\{ \left( \frac{U_M}{g_M^{-1}(\theta_1)} \right)^2 - \left( \frac{g_N^{-1}(\theta_j)}{g_N^{-1}(\theta_1)} \right)^2 \right\} +$$
$$\sum_{j=a_N+1}^{p+\alpha} \left\{ \left( \frac{U_M}{g_M^{-1}(\theta_1)} \right)^2 - \left( \frac{U_N}{g_N^{-1}(\theta_1)} \right)^2 \right\} + \sum_{j=1}^{b_M} \left\{ \left( \frac{g_M^{-1}(\theta_{p+q+1-j})}{g_M^{-1}(\theta_1)} \right)^2 - \left( \frac{g_N^{-1}(\theta_{p+q+1-j})}{g_N^{-1}(\theta_1)} \right)^2 \right\} +$$
$$\sum_{j=b_M+1}^{b_N} \left\{ \left( \frac{L_M}{g_M^{-1}(\theta_1)} \right)^2 - \left( \frac{g_N^{-1}(\theta_{p+q+1-j})}{g_N^{-1}(\theta_1)} \right)^2 \right\} + \sum_{j=b_N+1}^{q+\beta} \left\{ \left( \frac{L_M}{g_M^{-1}(\theta_1)} \right)^2 - \left( \frac{L_N}{g_N^{-1}(\theta_1)} \right)^2 \right\} \tag{31}$$

We have six summation terms in (31) and will show each term is negative.

(i) Consider the first term in (31):

$$S_1 = \sum_{j=2}^{a_M} \left\{ \left( \frac{g_M^{-1}(\theta_j)}{g_M^{-1}(\theta_1)} \right)^2 - \left( \frac{g_N^{-1}(\theta_j)}{g_N^{-1}(\theta_1)} \right)^2 \right\}$$

We will show each term $F_j = \left( \frac{g_M^{-1}(\theta_j)}{g_M^{-1}(\theta_1)} \right)^2 - \left( \frac{g_N^{-1}(\theta_j)}{g_N^{-1}(\theta_1)} \right)^2$ in the summation is negative. We can expand $F_j$ as follow:

$$
\begin{aligned}
F_j &= \left( \frac{g_M^{-1}(\theta_j)}{g_M^{-1}(\theta_1)} \right)^2 - \left( \frac{g_N^{-1}(\theta_j)}{g_N^{-1}(\theta_1)} \right)^2 \\
&= \left( \frac{g_M^{-1}(\theta_j)}{g_M^{-1}(\theta_1)} + \frac{g_N^{-1}(\theta_j)}{g_N^{-1}(\theta_1)} \right) \left( \frac{g_M^{-1}(\theta_j)}{g_M^{-1}(\theta_1)} - \frac{g_N^{-1}(\theta_j)}{g_N^{-1}(\theta_1)} \right) \\
&= \left( \frac{g_M^{-1}(\theta_j)}{g_M^{-1}(\theta_1)} + \frac{g_N^{-1}(\theta_j)}{g_N^{-1}(\theta_1)} \right) \left( \frac{g_M^{-1}(\theta_j)g_N^{-1}(\theta_1) - g_N^{-1}(\theta_j)g_M^{-1}(\theta_1)}{g_M^{-1}(\theta_1)g_N^{-1}(\theta_1)} \right)
\end{aligned} \tag{32}
$$

To verify the sign of $F_j$, we have to focus on the numerator of the second part of multiplicative term. We can simplify the numerator part as

$$g_M^{-1}(\theta_j)g_N^{-1}(\theta_1) - g_N^{-1}(\theta_j)g_M^{-1}(\theta_1)$$
$$= \theta_1 \left\{ s(N)\mathcal{R}_\mu(s(N)\theta_j^{-1}) - s(M)\mathcal{R}_\mu(s(M)\theta_j^{-1}) \right\} + \theta_j \left\{ s(M)\mathcal{R}_\mu(s(M)\theta_1^{-1}) - s(N)\mathcal{R}_\mu(s(N)\theta_1^{-1}) \right\}$$
$$+ s(N)s(M) \left\{ \mathcal{R}_\mu(s(M)\theta_1^{-1})\mathcal{R}_\mu(s(N)\theta_j^{-1}) - \mathcal{R}_\mu(s(N)\theta_1^{-1})\mathcal{R}_\mu(s(M)\theta_j^{-1}) \right\}$$

Consider the term:
$$F_{j,1} = \theta_1 \left\{ s(N)\mathcal{R}_\mu(s(N)\theta_j^{-1}) - s(M)\mathcal{R}_\mu(s(M)\theta_j^{-1}) \right\} + \theta_j \left\{ s(M)\mathcal{R}_\mu(s(M)\theta_1^{-1}) - s(N)\mathcal{R}_\mu(s(N)\theta_1^{-1}) \right\}$$

We know the $R$-transform can be expressed as power series as

$$\mathcal{R}_\mu(s(N)\theta^{-1}) = \sum_{n=1}^{\infty} C_n^{(\mu)} \left( \frac{s(N)}{\theta} \right)^{n-1}$$

where $C_n^{(\mu)}$ is the $n$-th cumulant of $\mu$. Then we can calculate $F_{j,1}$ as

$$F_{j,1} = \sum_{n=1}^{\infty} C_n^{(\mu)} (s(N)^n - s(M)^n) \, \theta_1 \cdot (1/\theta_j)^{n-1} - \sum_{n=1}^{\infty} C_n^{(\mu)} (s(N)^n - s(M)^n) \, \theta_j \cdot (1/\theta_1)^{n-1}$$
$$= \sum_{n=1}^{\infty} C_n^{(\mu)} (s(N)^n - s(M)^n) \left( \theta_1 \cdot \left( \frac{1}{\theta_j} \right)^{n-1} - \theta_j \cdot \left( \frac{1}{\theta_1} \right)^{n-1} \right)$$

Since $\theta_1 > \theta_j$ and $s(N) < s(M)$, we can easily show that $F_{j,1}$ is negative. Next, consider the term:

$$F_{j,2} = \mathcal{R}_\mu(s(N)\theta_j^{-1})\mathcal{R}_\mu(s(M)\theta_1^{-1}) - \mathcal{R}_\mu(s(M)\theta_j^{-1})\mathcal{R}_\mu(s(N)\theta_1^{-1})$$

By using the power series expression of $\mathcal{R}$-Transform, we have

$$F_{j,2} = \sum_{n=1}^{\infty} C_n^{(\mu)} \left( \frac{s(M)}{\theta_1} \right)^{n-1} \sum_{n=1}^{\infty} C_n^{(\mu)} \left( \frac{s(N)}{\theta_j} \right)^{n-1} - \sum_{n=1}^{\infty} C_n^{(\mu)} \left( \frac{s(N)}{\theta_1} \right)^{n-1} \sum_{n=1}^{\infty} C_n^{(\mu)} \left( \frac{s(M)}{\theta_j} \right)^{n-1}$$

We know that when $\sum_1^{\infty} a_n$ and $\sum_1^{\infty} b_N$ converges, then

$$\sum_1^{\infty} a_n \sum_1^{\infty} b_n = \sum_{n=1}^{\infty} \sum_{k=1}^{n} a_k b_{n-1+1}$$

Therefore, we have

$$F_{j,2} = \sum_{n=1}^{\infty} \sum_{k=1}^{n} C_k^{(\mu)} C_{n-k+1}^{(\mu)} \left( \frac{s(N)}{\theta_j} \right)^{k-1} \left( \frac{s(M)}{\theta_1} \right)^{n-k}$$
$$- \sum_{n=1}^{\infty} \sum_{k=1}^{n} C_k^{(\mu)} C_{n-k+1}^{(\mu)} \left( \frac{s(M)}{\theta_j} \right)^{k-1} \left( \frac{s(N)}{\theta_1} \right)^{n-k}$$
$$= \sum_{n=1}^{\infty} \sum_{k=1}^{n} C_k^{(\mu)} C_{n-k+1}^{(\mu)} \left\{ \left( \frac{s(N)}{\theta_j} \right)^{k-1} \left( \frac{s(M)}{\theta_1} \right)^{n-k} - \left( \frac{s(M)}{\theta_j} \right)^{k-1} \left( \frac{s(N)}{\theta_1} \right)^{n-k} \right\}$$

If we let $T(n) = \sum_{k=1}^{n} C_k^{(\mu)} C_{n-k+1}^{(\mu)} \left\{ \left( \frac{s(N)}{\theta_j} \right)^{k-1} \left( \frac{s(M)}{\theta_1} \right)^{n-k} - \left( \frac{s(M)}{\theta_j} \right)^{k-1} \left( \frac{s(N)}{\theta_1} \right)^{n-k} \right\}$, we can write $F_{j,2} = \sum_{n=1}^{\infty} T(n)$. We will show $F_{j,2}$ is negative by showing each $T(n)$ is negative for $n = 2m$ and $n = 2m + 1$, where $m \in \mathbb{N}$. For $n = 2m$ ($m \in \mathbb{N}$),

$$
\begin{aligned}
T(n) = T(2m) &= \sum_{k=1}^{2m} C_k^{(\mu)} C_{2m-k+1}^{(\mu)} \left\{ \left( \frac{s(N)}{\theta_j} \right)^{k-1} \left( \frac{s(M)}{\theta_1} \right)^{2m-k} - \left( \frac{s(M)}{\theta_j} \right)^{k-1} \left( \frac{s(N)}{\theta_1} \right)^{2m-k} \right\} \\
&= \sum_{k=1}^{m} \left[ C_k^{(\mu)} C_{2m-k+1}^{(\mu)} \left\{ \left( \frac{s(N)}{\theta_j} \right)^{k-1} \left( \frac{s(M)}{\theta_1} \right)^{2m-k} - \left( \frac{s(M)}{\theta_j} \right)^{k-1} \left( \frac{s(N)}{\theta_1} \right)^{2m-k} \right\} \right. \\
&\qquad \left. - C_{2m-k+1}^{(\mu)} C_k^{(\mu)} \left\{ \left( \frac{s(N)}{\theta_j} \right)^{2m-k} \left( \frac{s(M)}{\theta_1} \right)^{k-1} - \left( \frac{s(M)}{\theta_j} \right)^{2m-k} \left( \frac{s(N)}{\theta_1} \right)^{k-1} \right\} \right] \\
&= \sum_{k=1}^{m} C_k^{(\mu)} C_{2m-k+1}^{(\mu)} \left( s(N)^{k-1} s(M)^{2m-k} - s(M)^{k-1} s(N)^{2m-k} \right) \left( \frac{1}{\theta_j^{k-1} \theta_1^{2m-k}} - \frac{1}{\theta_j^{2m-k} \theta_1^{k-1}} \right)
\end{aligned}
$$

We can easily show two conditions:

$$
s(N)^{k-1} s(M)^{2m-k} - s(M)^{k-1} s(N)^{2m-k} > 0 \iff 2m - 2k + 1 > 0
$$

$$
\frac{1}{\theta_j^{k-1} \theta_1^{2m-k}} - \frac{1}{\theta_j^{2m-k} \theta_1^{k-1}} > 0 \iff 2m - 2k + 1 < 0
$$

Therefore, we can deduce $T(2m)$ is negative.

For $n = 2m + 1$ ($m \in \mathbb{N}$),

$$
T(n) = T(2m+1) = \sum_{k=1}^{2m+1} C_k^{(\mu)} C_{2m-k+2}^{(\mu)} \left\{ \left( \frac{s(N)}{\theta_j} \right)^{k-1} \left( \frac{s(M)}{\theta_1} \right)^{2m-k+1} - \left( \frac{s(M)}{\theta_j} \right)^{k-1} \left( \frac{s(N)}{\theta_1} \right)^{2m-k+1} \right\}
$$

$(m + 1)$-th term of $T(2m + 1)$ is zero since it is symmetric, thus we can write $T(2m + 1)$ as

$$
T(2m+1) = \sum_{k=1}^{m} C_k^{(\mu)} C_{2m-k+2}^{(\mu)} \left( s(N)^{k-1} s(M)^{2m-k+1} - s(M)^{k-1} s(N)^{2m-k+1} \right) \left( \frac{1}{\theta_j^{k-1} \theta_1^{2m-k+1}} - \frac{1}{\theta_j^{2m-k+1} \theta_1^{k-1}} \right)
$$

We can show $T(2m + 1)$ is negative as similar way of $T(2m)$. Therefore, $F_j$ is negative.

(ii) Consider the second term in (31):

$$
S_2 = \sum_{j=a_M+1}^{a_N} \left\{ \left( \frac{U_M}{g_M^{-1}(\theta_1)} \right)^2 - \left( \frac{g_N^{-1}(\theta_j)}{g_N^{-1}(\theta_1)} \right)^2 \right\}
$$

We will show $S_2$ is negative by showing each term in the summation $G_j = \frac{U_M^2}{\left\{ \theta_1 + s(M) \mathcal{R}_\mu(s(M) \theta_1^{-1}) \right\}^2} - \left\{ \frac{\theta_j + s(N) \mathcal{R}_\mu(s(N) \theta_j^{-1})}{\theta_1 + s(N) \mathcal{R}_\mu(s(N) \theta_1^{-1})} \right\}^2$ is negative. Since $\theta_j + s(M) \mathcal{R}_\mu(S(M) \theta_j^{-1}) > U_M$ for all $j \in \{a_M + 1, \cdots, a_N\}$, we have $G_j < F_j < 0$. Therefore, $S_2$ is negative.

(iii) Consider the third term in (31):

$$S_3 = \sum_{j=a_N+1}^{p+\alpha} \left\{ \left( \frac{U_M}{g_M^{-1}(\theta_1)} \right)^2 - \left( \frac{U_N}{g_N^{-1}(\theta_1)} \right)^2 \right\}$$

$$= (p + \alpha - a_N - 1) \left( \frac{U_M^2}{\left\{ \theta_1 + s(M)\mathcal{R}_\mu(s(M)\theta_1^{-1} \right\}^2} - \frac{U_N^2}{\left\{ \theta_1 + s(N)\mathcal{R}_\mu(s(N)\theta_1^{-1} \right\}^2} \right)$$

By using the fact $U_N/s(N) = U_M/s(M)$, we can write the above term as

$$S_3 = (p + \alpha - a_N - 1) \left( \frac{U_M^2}{\left\{ \theta_1 + s(M)\mathcal{R}_\mu(s(M)\theta_1^{-1} \right\}^2} - \frac{U_M^2}{\left\{ \theta_1 + s(N)\mathcal{R}_\mu(s(N)\theta_1^{-1} \right\}^2} \frac{s(N)^2}{s(M)^2} \right)$$

By using the power series expansion of $\mathcal{R}$-Transform, we can easily show that $Q = 0$. For fourth, fifth, and sixth terms in (31), they are negative in similar way of (i), (ii), and (iii). Therefore, $\hat{\text{srank}}(H_M) - \hat{\text{srank}}(H_N)$ is negative.

$\square$

## B    DETAIL OF THE ALGORITHMS

In this section, we provide a detailed explanation of personalized version and model-heterogeneous version , including the datasets and hyperparameters used. The implementation is based on PyTorch.

### B.1    PERSONALIZED FEDERATED LOW-RANK UPDATES (pFedLoRU)

---

**Algorithm 2** pFedLoRU. $W$ is a model, $A_0, B_0$ are initial global low-rank update matrices, $L_0, U_0$ are initial personal low-rank update matrices, $\alpha_{\text{global}}, \alpha_{\text{per}}$ are the scaling factors, $\tau$ is an accumulation cycle, $T$ is the total training round

---
**Require:** $W, L_0, U_0, A_0, B_0, \alpha_{\text{global}}, \alpha_{\text{per}}, \tau, T$
  **Initialize:** Server sends $W$ to each client.
  **for** $t = 1, \cdots, T$ **do**
      Server selects $M$ clients $\mathcal{K}_M$ and distributes $A_{t-1}, B_{t-1}$ to the clients in $\mathcal{K}_M$.
      **for** each client $k \in \mathcal{K}_M$ **do**
         **Local training:**
         Find $L_t^{(k)}, U_t^{(k)}$ by solving (7) starting from $W + \alpha_{\text{global}} A_{t-1} B_{t-1} + \alpha_{\text{per}} L_{t-1}^{(k)} U_{t-1}^{(k)}$.
         Find $A_t^{(k)}, B_t^{(k)}$ by solving (8) starting from $W + \alpha_{\text{global}} A_{t-1} B_{t-1} + \alpha_{\text{per}} L_t^{(k)} U_t^{(k)}$.
         Send $A_t^{(k)}, B_t^{(k)}$ to the server.
      **end for**
      **Server aggregation:** $A_t \leftarrow \sum_{k \in \mathcal{K}_M} p^{(k)} A_t^{(k)}, B_t \leftarrow \sum_{k \in \mathcal{K}_M} p^{(k)} B_t^{(k)}$.
      **if** $t \bmod \tau = 0$ **then**
         Server distributes $A_t, B_t$ to all clients .
         Each client $k$ updates its local copy of the global model: $W \leftarrow W + \alpha_{\text{global}} A_t B_t$
      **end if**
  **end for**
  **Return:** $W + \sum_{t=1:\, t \bmod \tau = 0}^{T} A_t B_t + L_T^{(k)} U_T^{(k)}$ for all client $k$

---

The pFedLoRU algorithm enables each client $k$ to train a personalized model adapted to its data distribution. In pFedLoRU, client $k$ retains global low-rank update matrices $A^{(k)}$ and $B^{(k)}$ for updating the shared model, as well as personalized low-rank update matrices $L^{(k)}$ and $U^{(k)}$ for

learning the personalized model. The communication between the server and clients involves only the low-rank matrices $\boldsymbol{A}^{(k)}$ and $\boldsymbol{B}^{(k)}$, which substantially reduces communication overhead. These matrices, $\boldsymbol{A}^{(k)}$ and $\boldsymbol{B}^{(k)}$, are aggregated to update the local copy of the global model $\boldsymbol{W}$. Finally, each client possesses a personalized model of the form $\boldsymbol{W} + \boldsymbol{L}^{(k)}\boldsymbol{U}^{(k)}$.

In practice, since the global model incorporates general knowledge from the all clients' dataset, and the personalized model is essentially a fine-tuned version of the global model, we typically assign higher ranks to $\boldsymbol{A}^{(k)}$ and $\boldsymbol{B}^{(k)}$. Additionally, although we use the same rank for $\boldsymbol{L}^{(k)}$ and $\boldsymbol{U}^{(k)}$ across all clients in our experiments, each client can, in practice, use different ranks based on the complexity and size of their local dataset. It is also noteworthy that different ranks for $\boldsymbol{A}^{(k)}$ and $\boldsymbol{B}^{(k)}$ can be employed by integrating pFedLoRU and mFedLoRU.

## B.2 Model-Heterogeneous Federated Low-Rank Updates (mFedLoRU)

---

**Algorithm 3** mFedLoRU. $\boldsymbol{W}$ is a model, $\boldsymbol{A}_0, \boldsymbol{B}_0$ are initial low-rank update matrices, $\alpha, \alpha_A^{(k)}, \alpha_B^{(k)}$ are scaling factors, $\tau$ is an accumulation cycle, $T$ is the total training round.

---

**Require:** $\boldsymbol{W}, \boldsymbol{A}_0, \boldsymbol{B}_0, \alpha, \alpha_A^{(k)}, \alpha_B^{(k)}, \tau, T$
  **Initialize:** Server sends $\boldsymbol{W}$ to each client.
  **for** $t = 1, \cdots, T$ **do**
    Server selects $M$ clients $\mathcal{K}_M$ and distributes $\boldsymbol{A}_{t-1}, \boldsymbol{B}_{t-1}$.
    **for** each client $k \in \mathcal{K}_M$ **do**
      Initializes nested low-rank updates $\boldsymbol{A}_{\mathrm{d}}^{(k)}, \boldsymbol{A}_{\mathrm{u}}^{(k)}$ and $\boldsymbol{B}_{\mathrm{d}}^{(k)}, \boldsymbol{B}_{\mathrm{u}}^{(k)}$.
      **Local training:**
      Find $\boldsymbol{A}_{\mathrm{d}}^{(k)}, \boldsymbol{A}_{\mathrm{u}}^{(k)}, \boldsymbol{B}_{\mathrm{d}}^{(k)}, \boldsymbol{B}_{\mathrm{u}}^{(k)}$ by solving (9)
      starting from $\boldsymbol{W} + \alpha(\boldsymbol{A}_{t-1} + \alpha_A^{(k)}\boldsymbol{A}_{\mathrm{d}}^{(k)}\boldsymbol{A}_{\mathrm{u}}^{(k)})(\boldsymbol{B}_{t-1} + \alpha_B^{(k)}\boldsymbol{B}_{\mathrm{d}}^{(k)}\boldsymbol{B}_{\mathrm{u}}^{(k)})$.
      Sends $\boldsymbol{A}_{\mathrm{d}}^{(k)}\boldsymbol{A}_{\mathrm{u}}^{(k)}$ and $\boldsymbol{B}_{\mathrm{d}}^{(k)}\boldsymbol{B}_{\mathrm{u}}^{(k)}$ to the server.
    **end for**
    Recover rank-$r$ low-rank updates from hierarchical low-rank updates:
    $\boldsymbol{A}_t^{(k)} \leftarrow \boldsymbol{A}_{t-1} + \alpha_A^{(k)}\boldsymbol{A}_{\mathrm{d}}^{(k)}\boldsymbol{A}_{\mathrm{u}}^{(k)}, \quad \boldsymbol{B}_t^{(k)} \leftarrow \boldsymbol{B}_{t-1} + \alpha_B^{(k)}\boldsymbol{B}_{\mathrm{d}}^{(k)}\boldsymbol{B}_{\mathrm{u}}^{(k)}$.
    **Server aggregation:** $\boldsymbol{A}_t \leftarrow \sum_{k \in \mathcal{K}_M} p^{(k)}\boldsymbol{A}_t^{(k)}, \boldsymbol{B}_t \leftarrow \sum_{k \in \mathcal{K}_M} p^{(k)}\boldsymbol{B}_t^{(k)}$.
    **if** $t \bmod \tau = 0$ **then**
      Server distributes $\boldsymbol{A}_t, \boldsymbol{B}_t$ to all clients.
      Each client $k$ updates its local model: $\boldsymbol{W} \leftarrow \boldsymbol{W} + \alpha\boldsymbol{A}_t\boldsymbol{B}_t$.
    **end if**
  **end for**
  **Return:** $\boldsymbol{W} + \sum_{t=1:\, t \bmod \tau = 0}^{T} \boldsymbol{A}_t\boldsymbol{B}_t$.

---

Model-heterogeneous FedLoRU (mFedLoRU) algorithm enables each client $k$ to utilize a rank tailored to its resource constraints. Similar to FedLoRU, client $k$ maintains low-rank update matrices $\boldsymbol{A}^{(k)} \in \mathbb{R}^{m \times r}$ and $\boldsymbol{B}^{(k)} \in \mathbb{R}^{r \times n}$, but employs recursive low-rank updates during training. Each client $k$ decides whether to use nested low-rank updates or not. If a client opts out of nested low-rank updates, it updates its low-rank modules like in FedLoRU. However, if client $k$ chooses nested low-rank updates, it determines the locally adapted rank $r_A^{(k)}, r_B^{(k)} < r$ based on its resources. At each round, it initializes nested low-rank update matrices $\boldsymbol{A}_{\mathrm{d}}^{(k)} \in \mathbb{R}^{m \times r_A^{(k)}}, \boldsymbol{A}_{\mathrm{u}}^{(k)} \in \mathbb{R}^{r_A^{(k)} \times r}$ and $\boldsymbol{B}_{\mathrm{d}}^{(k)} \in \mathbb{R}^{r \times r_B^{(k)}}, \boldsymbol{B}_{\mathrm{u}}^{(k)} \in \mathbb{R}^{r_B^{(k)} \times n}$ such that $\boldsymbol{A}_{\mathrm{d}}^{(k)}\boldsymbol{A}_{\mathrm{u}}^{(k)} = 0$ and $\boldsymbol{B}_{\mathrm{d}}^{(k)}\boldsymbol{B}_{\mathrm{u}}^{(k)} = 0$. After local training by solving (9), we can update client $k$'s original low-rank matrices as follows:

$$\boldsymbol{A}^{(k)} \leftarrow \boldsymbol{A}^{(k)} + \alpha_A^{(k)}\boldsymbol{A}_{\mathrm{d}}^{(k)}\boldsymbol{A}_{\mathrm{u}}^{(k)}, \quad \boldsymbol{B}^{(k)} \leftarrow \boldsymbol{B}^{(k)} + \alpha_A^{(k)}\boldsymbol{B}_{\mathrm{d}}^{(k)}\boldsymbol{B}_{\mathrm{u}}^{(k)} \tag{33}$$

After local training, to reduce communication overhead, the client does not recover its original low-rank matrices directly. Instead, it sends the nested low-rank matrices to the server, which recovers them into rank-$r$ low-rank matrices $\boldsymbol{A}^{(k)} \leftarrow \boldsymbol{A} + \alpha_A^{(k)}\boldsymbol{A}_{\mathrm{d}}^{(k)}\boldsymbol{A}_{\mathrm{u}}^{(k)}$, and $\boldsymbol{B}^{(k)} \leftarrow \boldsymbol{B} + \alpha_B^{(k)}\boldsymbol{B}_{\mathrm{d}}^{(k)}\boldsymbol{B}_{\mathrm{u}}^{(k)}$, and then performs aggregation using these rank-$r$ low-rank matrices as in FedLoRU. By using this strategy, the communication overhead is reduced from $2mn$ to $r(m+n) + r_A(m+r) + r_B(n+r)$.

## B.3 Personalized Federated Low-Rank Adaptation (pFedLoRA)

---

**Algorithm 4** pFedLoRA. $W$ is a model, $L_0, U_0$ are initial personal low-rank update matrices, $\alpha_{\text{per}}$ is the scaling factor, $T$ is the total training round.

---

**Require:** $W$, $L_0$, $U_0$, $\alpha_{\text{per}}$, $T$.
    **for** $t = 1, \cdots, T$ **do**
        Server selects $M$ clients $\mathcal{K}_M$ and distributes $W_{t-1}$ and client $k$ initializes it
        as a local copy of the global model.
        **for** each client $k \in \mathcal{K}_M$ **do**
            **Local training - pFedLoRA(1):**
            Find $L_t^{(k)}, U_t^{(k)}$ by solving (34) starting from $W_{t-1} + \alpha_{\text{per}} L_{t-1}^{(k)} U_{t-1}^{(k)}$.
            Find $W_t^{(k)}$ by solving (35) starting from $W_{t-1} + \alpha_{\text{per}} L_t^{(k)} U_t^{(k)}$.
            **Local training - pFedLoRA(2):**
            Find $W_t^{(k)}, L_t^{(k)}, U_t^{(k)}$ together by solving (36) starting from $W_{t-1} + \alpha_{\text{per}} L_{t-1}^{(k)} U_t^{(k)}$.
            Send $W_t^{(k)}$ to the server.
        **end for**
        **Server aggregation:** $W_t \leftarrow \sum_{k \in \mathcal{K}_M} p^{(k)} W_t^{(k)}$.
    **end for**
    **Return:** $W_T + L_T^{(k)} U_T^{(k)}$ for all client $k$.

---

We outline two variants of the personalized FedLoRA algorithm here. Both versions of pFedLoRA follow a similar framework, where each client maintains a full-rank global model $W$ and its own personalization modules $L^{(k)}$ and $U^{(k)}$.

In pFedLoRA(1), the first variant, as suggested by Wu et al. (2024) and other FedLoRA algorithms, the personalization modules are optimized separately from the global model. Specifically, the algorithm first optimizes the personalization modules for $E_{\text{per}}$ and subsequently optimizes the global full-rank model for $E_{\text{global}}$ by solving:

$$L_t^{(k)}, U_t^{(k)} = \arg\min_{L, U} f^{(k)}(W_{t-1} + \alpha_{\text{per}} LU) \tag{34}$$

$$W_t^{(k)} = \arg\min_{W} f^{(k)}(W + \alpha_{\text{per}} L_t^{(k)} U_t^{(k)}) \tag{35}$$

However, pFedLoRA(1) has been found to be less effective compared to our modified version pFedLoRA(2). The second variant, pFedLoRA(2), optimizes both the personalization modules and the global full-rank model simultaneously for $E = E_{\text{per}} + E_{\text{global}}$ by solving:

$$W_t^{(k)}, L_t^{(k)}, U_t^{(k)} = \arg\min_{W, L, U} f^{(k)}(W + \alpha_{\text{per}} LU) \tag{36}$$

## C Detail of the experiment setting

In this section, we provide a detailed explanation of the experiments, including the datasets and hyperparameters used. The implementation is based on PyTorch.

### C.1 Datasets and Models

The federated learning experiments were performed using four datasets: Fashion-MNIST (FMNIST, Xiao et al. (2017)), CIFAR-10, CIFAR-100 (Krizhevsky et al., 2009), and Alpaca (Taori et al., 2023). Detailed statistics for these datasets are provided in Table A1. The Alpaca dataset, consisting of 52,000 instruction and demonstration samples, was divided into 50,000 instances for training and 2,000 for testing in our fine-tuning experiment.

Table A1: Description of datasets used in the experiments

| Dataset | Number of Classes | Total Samples | | Samples per class | |
|---|---|---|---|---|---|
| | | Training | Validation | Training | Validation |
| **FMNIST** | 10 | 60000 | 10000 | 6000 | 1000 |
| **CIFAR-10** | 10 | 50000 | 10000 | 5000 | 1000 |
| **CIFAR-100** | 100 | 50000 | 10000 | 500 | 100 |
| **Alpaca** | - | 50000 | 2000 | - | - |

We construct datasets for clients by evenly splitting the training data among $K$ clients in a statistically homogeneous (i.e., iid) federated learning setting. For the heterogeneous statistical setting, we follow the procedure outlined in Hsu et al. (2019), which involves applying latent Dirichlet allocation (LDA) over the dataset labels to create clients' datasets. In this approach, each client is assigned a multinomial distribution over the labels, from which its examples are sampled. The multinomial distribution is drawn from a symmetric Dirichlet distribution with parameter $\psi$. For the non-iid setting, we use $\psi = 0.5$ to simulate a severely heterogeneous environment.

Table A2: ResNet-10 and ResNet-18 architecture for image classification datasets.

| Layer Name | ResNet-10 | ResNet-18 |
|---|---|---|
| conv1 | $3\times3$, 64, stride 1, padding 1 | $3\times3$, 64, stride 1, padding 1 |
| layer1 | $\begin{bmatrix} 3 \times 3, 64 \\ 3 \times 3, 64 \end{bmatrix} \times 1$ | $\begin{bmatrix} 3 \times 3, 64 \\ 3 \times 3, 64 \end{bmatrix} \times 2$ |
| layer2 | $\begin{bmatrix} 3 \times 3, 128 \\ 3 \times 3, 128 \end{bmatrix} \times 1$ | $\begin{bmatrix} 3 \times 3, 128 \\ 3 \times 3, 128 \end{bmatrix} \times 2$ |
| layer3 | $\begin{bmatrix} 3 \times 3, 256 \\ 3 \times 3, 256 \end{bmatrix} \times 1$ | $\begin{bmatrix} 3 \times 3, 256 \\ 3 \times 3, 256 \end{bmatrix} \times 2$ |
| layer4 | $\begin{bmatrix} 3 \times 3, 512 \\ 3 \times 3, 512 \end{bmatrix} \times 1$ | $\begin{bmatrix} 3 \times 3, 512 \\ 3 \times 3, 512 \end{bmatrix} \times 2$ |

Table A2 illustrates the model architectures used in the experiments on Fashion MNIST (FMNIST), CIFAR-10, and CIFAR-100. We employ ResNet-10 for FMNIST and ResNet-18 (Krizhevsky et al., 2009) for CIFAR-10 and CIFAR-100. Each ResNet model includes a fully connected layer at the end, and the total number of model parameters varies slightly depending on the number of classes in the dataset. The parameter counts for the original models are as follows: ResNet-10 with 10 classes has 4.90M parameters; ResNet-18 with 10 classes has 11.17M parameters; and ResNet-18 with 100 classes has 11.22M parameters. For fine-tuning on Alpaca, we utilize the pre-trained LLaMA2-3B model (Touvron et al., 2023).

## C.2 IMPLEMENTATION AND TRAINING DETAILS

**Detailed implementation of FedLoRA, FedLoRU, and FedHM** In FedLoRA, FedLoRU, FedHM, and their variant algorithms, we apply low-rank factorization to the convolutional layers in ResNet-based models and to the self-attention modules in LLaMA2-3B. Specifically, for ResNet10 and ResNet18, we factorize the convolutional layers in layer1 through layer4, and for LLaMA2-3B, we factorize the self-attention modules in q_proj, k_proj, v_proj, and o_proj. We explore various low-rank configurations, setting the ranks of the factorized modules to 16, 32, 64, and 128 for FedLoRA and FedLoRU. We use rank $r = 128$ as the largest rank since our initial experiments showed it to have the best performance/memory trade-off. For FedHM, since its factorization scheme differs from that of FedLoRA and FedLoRU, we determine equivalent rank factors that yield the same number of trainable parameters as the ranks used in FedLoRA and FedLoRU.

We employ two strategies for initializing the low-rank update matrices in FedLoRU. For random initialization, as adopted in Hu et al. (2021), we initialize $A$ with a random Gaussian distribution and set $B$ to zero, ensuring that $AB$ is zero at the start. Alternatively, for momentum initialization, we retain the existing weights of the matrices, continuing to use the previous low-rank update matrices. This approach leverages momentum effects as described in the ReLoRA(Lialin et al., 2023). The

scheduling of accumulations is also critical due to the varying nature of the training phases across different rounds; in this study, we employ periodic accumulation with the accumulation cycle determined through a grid search over the values {20, 30, 40, 50, 60, 70, 80}, though this area warrants further investigation. We assess the performance by evaluating Top-1 test accuracy across experiments. In the non-iid setting, due to significant fluctuations in performance, we report the average of the last five test accuracy values.

**Federated learning setting**   The federated learning experiments were conducted using four datasets: FMNIST, CIFAR-10, CIFAR-100, and Alpaca. The client sampling rate, representing the proportion of clients selected per communication round, was set at 0.5 for all datasets. Each client performed 5 local epochs per communication round on the image datasets with a batch size of 32, while client performed 1 local epochs on Alpaca with a batch size of 16.

For training FMNIST, CIFAR-10, and CIFAR-100, we utilized stochastic gradient descent (SGD) with a momentum of 0.9 as the local optimizer. The learning rate was selected through a grid search over 0.3, 0.2, 0.1, 0.05, 0.01, and a Cosine-Annealing learning rate scheduler was applied throughout the training process, with a minimum learning rate of 0.001 and a cycle step set to 50 or the total number of communication rounds. For fine-tuning LLaMA2-3B, we used AdamW (Loshchilov, 2017) as the local optimizer, with a learning rate of 3e-4 and betas set to (0.9, 0.999), without employing a learning rate scheduler.

**Fine-tuning setting**   We assess the fine-tuning performance of FedLoRA and FedLoRU using two different ranks, 8 and 16. For the low-rank matrix factorization of LLaMA2-3B, we employ the PEFT library (Mangrulkar et al., 2022). The percentage of trainable parameters is 0.124% for rank 8 and 0.248% for rank 16.

**Model heterogeneous setting**   Here we describe the model heterogeneous settings used in our experiments. To simulate varying client capabilities, we tested two different model heterogeneous configurations in mFedLoRU experiments where the clients had different ranks, denoted as $r$, which reflect the computational resources or constraints of each client. For FedHM, we match the number of trainable parameters corresponding to the model with specific rank in mFedLoRU experiments.

Table A3: Detailed model heterogeneous settings in our experiments. Both settings include total 20 clients.

| **Rank of a client** | | $r = 128$ | $r = 64$ | $r = 32$ | $r = 16$ |
|---|---|---|---|---|---|
| **#Clients** | **setting 1** | 5 | 5 | 5 | 5 |
| | **setting 2** | - | 6 | 6 | 7 |

The motivation behind these settings was to establish a challenging model heterogeneous environment. This is particularly important as we observed that FedLoRU with $r = 128$ produces similar results to FedAvg with a full-rank model. Therefore, these configurations were designed to test the algorithm's adaptability under more demanding and diverse client conditions. In addition, we set $\alpha_A$ and $\alpha_B$ to satisfy $\alpha_A/r_A = \alpha_A/r_B = 1/2$, as our empirical observations indicate that the choice of $\alpha$ values in the range of $1/4$ to $1$ has minimal effect on overall performance.

C.3   DETAIL OF THE ESTIMATED STABLE RANK EXPERIMENT

We conduct an experiment to support our theoretical analysis that the Hessians of loss functions trained on smaller datasets exhibit larger stable ranks. In this experiment, we randomly select either 50 or 500 samples from the CIFAR-100 dataset and train a ResNet-18 model using only these 50 or 500 samples. Every 5 epochs, we compute an estimated stable rank of the Hessian, as calculating the true stable rank is computationally challenging due to the need to determine all singular values. Instead, we estimate the empirical spectral density using pyhessian (Yao et al., 2020), which provides the empirical singular values $\sigma_i(H)$ of a Hessian $H$ and their corresponding densities $p(\sigma_i)$, $i = 1, \cdots, Q$. Based on this, we calculate the estimated stable rank as follows:

$$\hat{\text{srank}}(H) = \frac{\sum_{i=1}^{Q} p(\sigma_i) \, \sigma_i^2(H)}{p(\sigma_1) \, \sigma_1^2(H)} \tag{37}$$

Figure 2(b) shows the results of the experiment, demonstrating that the Hessians trained on the smaller dataset ($n = 50$) consistently exhibits higher estimated stable ranks compared to those trained on the larger dataset ($n = 500$).

## D  FURTHER DISCUSSION ON EXPERIMENT RESULTS

In this section, we present learning curve plots and additional experimental results that were not included in the main text. Furthermore, we provide a more detailed analysis and discussion of the experimental outcomes.

### D.1  EXPERIMENT RESULTS FOR FEDAVG

To emphasize the comparison between FedLoRU and other communication-efficient federated learning algorithms, we have excluded the FedAvg experiment results from the main text. The FedAvg outcomes are instead provided in Table A4.

Table A4: Top-1 test accuracy of FedAvg under different federated learning settings and datasets

|  | Dataset | FMNIST | CIFAR-10 | CIFAR-100 |
|---|---|---|---|---|
|  | **IID - K=20** | 91.81 | 93.48 | 69.97 |
| **FL setting** | **IID - K=100** | 90.19 | 85.14 | 55.14 |
|  | **NonIID - K=20** | 80.03 | 79.65 | 19.18 |

From Table 1 and Table A4, we observe that FedAvg consistently performs well across different datasets and settings, but its performance tends to drop as the number of clients increases and in non-IID scenarios. For example, in the CIFAR-100 dataset under the IID setting with 100 clients, FedAvg achieves a test accuracy of 55.14%, while its accuracy drops significantly to 19.18% in the non-IID setting with 20 clients. This illustrates FedAvg's limitations in handling large client numbers and heterogeneous data distributions.

In comparison, FedLoRU demonstrates competitive performance relative to FedAvg. While Fed-LoRU is at most 5% less accurate than FedAvg in some cases, it sometimes outperforms FedAvg, particularly in scenarios with a larger number of clients. For instance, in the CIFAR-100 IID setting with 100 clients, FedLoRU achieves a test accuracy of 57.96%, which surpasses FedAvg's accuracy of 55.14%. This suggests that FedLoRU's low-rank update approach scales better with an increasing number of clients and is more robust in large-scale federated learning environments.

### D.2  LEARNING CURVE PLOTS FOR IID SETTING

We present the test accuracy curves for experiments conducted under a statistically homogeneous setting. Figure A1 and Figure A2 shows the test accuracy w.r.t. communication round under iid setting. The fluctuations observed in the graphs are attributable to the use of a cosine-annealing learning rate scheduler.

### D.3  DISCUSSION ON COMMUNICATION COST

One of the main motivation of FedLoRU is to reduce the communication cost by using low-rank updates while maintaining reasonable performances. When the original weight matrix $W \in \mathbb{R}^{m \times n}$ requires $mn$ parameters to be communicated, FedLoRU with rank $r$ requires $r(m + n)$ parameters. Additionally, as we can see in Figure A1 and Figure A2, convergence speed is similar to FedAvg, resulting much lower communication overheads.

Building on the motivation to reduce communication costs, Figure 3(b) compares the communication overheads across several federated learning algorithms—FedAvg, FedHM, FedLoRA, and

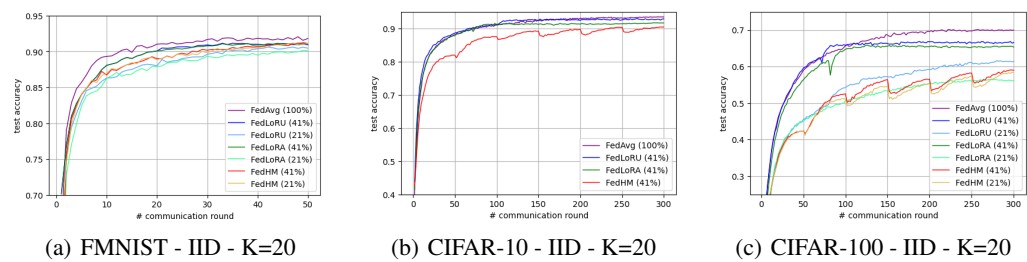

(a) FMNIST - IID - K=20     (b) CIFAR-10 - IID - K=20     (c) CIFAR-100 - IID - K=20

Figure A1: The test accuracy v.s. communication round under IID and K=20 setting.

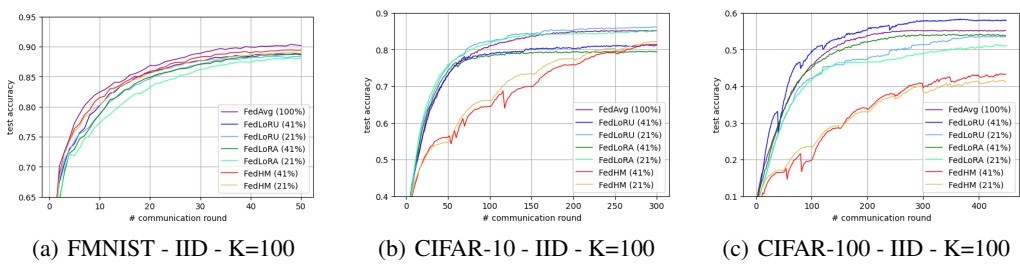

(a) FMNIST - IID - K=100     (b) CIFAR-10 - IID - K=100     (c) CIFAR-100 - IID - K=100

Figure A2: The test accuracy v.s. communication round under IID and K=100 setting.

FedLoRU—using the CIFAR-10 and CIFAR-100 datasets. The figure evaluates the communication cost in gigabytes (GB) required to reach specific target test accuracy (denoted as $T\%$) for different numbers of clients ($K$) and datasets. We compute the communication cost as $2 \times$ (#clients) $\times$ (participation rate) $\times$ (#parameters) $\times$ (parameter memory size) $\times$ (#round). It is evident that FedLoRU consistently achieves significantly lower communication costs compared to the other methods.

### D.4 RELATIVE DIFFERENCE IN PERFORMANCE IN TERMS OF THE NUMBER OF CLIENTS

Table A5 presents a comparison of test accuracy between FedAvg, FedLoRA, and FedLoRU across varying client numbers, illustrating the relative performance of these algorithms as the number of clients increases. FedLoRU consistently outperforms FedAvg when the number of clients exceeds 100, demonstrating its scalability and effectiveness in cross-device federated learning environments. Interestingly, even FedLoRA, which does not accumulate low-rank updates as in FedLoRU, outperforms FedAvg, particularly when the number of clients reaches 200 and above. This result suggests that simply adopting low-rank updates in cross-device FL can significantly improve performance. These findings align with our theoretical insights, highlighting the potential benefits of leveraging low-rank structures in federated learning, even without the accumulation strategy employed by FedLoRU.

Table A5: A comparison between FedAvg, FedLoRA, and FedLoRU accuracy across varying client numbers. The ratio is the relative difference in accuracy between two algorithms. Here, we compute the ratio of FedLoRA and FedLoRU compared to FedAvg. For example, ratio of FedLoRU is defined as Ratio = $\frac{\text{FedLoRU} - \text{FedAvg}}{\text{FedLoRU}}$.

| #Clients | FedAvg | FedLoRA acc | FedLoRA ratio | FedLoRU acc | FedLoRU ratio |
|---|---|---|---|---|---|
| 20 | 69.97 | 65.53 | -0.063 | 66.81 | -0.046 |
| 50 | 64.68 | 59.87 | -0.074 | 62.45 | -0.034 |
| 100 | 55.14 | 53.79 | -0.024 | 57.96 | +0.051 |
| 200 | 38.85 | 42.42 | +0.092 | 44.85 | +0.154 |
| 300 | 24.94 | 32.69 | +0.311 | 36.79 | +0.475 |
| 400 | 21.44 | 31.41 | +0.465 | 35.86 | +0.673 |