# OpenReview forum: "Communication-Efficient Federated Low-Rank Update Algorithm and its Connection to Implicit Regularization"
_ICLR.cc/2025/Conference — Submitted to ICLR 2025_

### Official Review · Reviewer_yeeh · 2024-10-27

**Soundness:** 2
**Presentation:** 3
**Contribution:** 2
**Rating:** 5
**Confidence:** 3

**Summary:**

This paper studies communication-efficient low-rank update framework for federated learning.

It provides theoretical asymptotic analysis for the rank structures of the Hessian at server side and client side, which motivates the design of FedLoRU algorithm. Generalizations of FedLoRU under statistical and model heterogeneity, namely pFedLoRU and mFedLoRU, are also presented. Finally, the authors conduct experiments on computer vision pre-training and language model fine-tuning tasks to demonstrate the performance of FedLoRU and its generalizations.

**Strengths:**

* The paper provides rigorous theoretical analysis on the Hessian rank structures, establishing interesting asymptotic results within a mathematically general framework.
* The proposed algorithm achieves performance comparable or superior to other known methods in experiments, while significantly reducing the communication overhead by low-rank updates.
* The presentation of this paper is well-organized and the motivation and methodology are clear to follow.

**Weaknesses:**

* Although the authors provide some Hessian rank structure analysis, the design of FedLoRU can be better supported from the theoretical side. For example, some convergence guarantees, since low-rank updates lead to loss of information compared to full-rank updates and may hurt the optimization.
* The title mentions "its connection to implicit regularization", but I was not able to spot sufficient discussion on implicit regularization of FedLoRU; also please see a conceptual question below.
* The design of FedLoRU seems a straightforward extension to federated setting of existing methods for low-rank matrix accumulation such as ReLoRA [1].
* This is not a major weakness but more evaluations on LLM fine-tuning could be done, as most of the experiment details are devoted to computer vision tasks on small datasets.

[1] Vladislav Lialin, Sherin Muckatira, Namrata Shivagunde, and Anna Rumshisky. Relora: High- rank training through low-rank updates. In The Twelfth International Conference on Learning Representations, 2023.

**Questions:**

* The title mentions "its connection to implicit regularization". To my knowledge, implicit regularization refers to the phenomenon that optimizers without explicit regularization, such as SGD, prefer regularized solutions [1]. However, FedLoRU explicitly works in a specific rank-$r$ space. Could the authors please explain in what sense is FedLoRU connected to implicit regularization?
* The theory part analyzes the rank structures of *loss Hessians* at server and client side. At the same time, FedLoRU proposes to perform low-rank updates on the model's *weight matrices*. Could the authors please explain the connection between the rank structure of loss Hessians and weight matrices?

[1] Ziwei Ji and Matus Telgarsky. Gradient descent aligns the layers of deep linear networks. arXiv preprint arXiv:1810.02032, 2018.

---

> ### Author Response · Authors · 2024-11-19
> **Response to Reviewer yeeh's Review**
>
> We appreciate the reviewer’s time. Below, we address the concerns and questions raised:
>
> **Q1. Could the authors please explain in what sense is FedLoRU connected to implicit regularization?**
>
> Thank you for the thoughtful question. We would like to clarify how FedLoRU is connected to implicit regularization.
>
> Mathematically, regularization refers to the process of simplifying the solution of a problem. Explicit regularization involves adding a penalty term to the optimization objective, whereas implicit regularization includes all forms of regularization that do not involve such penalties. Since FedLoRU does not introduce any explicit penalty term, its regularization effect is implicit.
>
> Now I will explain the implicit regularization effect of FedLoRU.
>
> In federated learning, a key challenge is the discrepancy between clients. After local training, individual models often diverge significantly, leading to suboptimal performance when aggregated at the server. Reducing this client discrepancy is crucial for improving federated learning outcomes.
>
> Our theoretical analysis (Theorem 3.2) shows that clients exhibit a higher stable rank, indicating a more complex loss landscape. This complexity exacerbates client discrepancies. By constraining updates to a low-rank space, FedLoRU implicitly regularizes client training, aligning updates along major directions and reducing client-to-client variations. In short, by using local low-rank updates, we force clients to have simpler solutions (locally trained models) which leads to more general aggregated server model.
>
>
> **Q2. Could the authors please explain the connection between the rank structure of loss Hessians and weight matrices?**
>
> We would like to clarify the connection between the rank structure of loss Hessians and the model's weight matrices in the context of FedLoRU.
>
> 1. **Role of Low-Rank Matrices:**
>
>     In FedLoRU, low-rank matrices are used to represent the updates applied during local training. These matrices capture the difference between the updated model and the previous model. The connection between the Hessian and the model lies in the fact that model updates follow the eigenvectors of the Hessian. This relationship is critical to understanding the curvature of the loss landscape and its effect on training.
>
> 2. **Connection Through Loss Landscape Analysis:**
>
>     In our paper, we use **stable rank** to analyze the rank properties of the loss landscape at both the client and server levels. The rank nature reflects the curvature information of the loss landscape, which directly influences the model updates during training. Specifically, stable rank provides a better representation of this curvature compared to rank, as it focuses on the effective dimensionality of the Hessian.
>
> 3. **Insights from Deep Learning Training Dynamics**:
>
>     According to the works ([1], [2], [3]), gradient descent trajectory is separated into two components: a bulk component (eigenvectors corresponding to large number of small eigenvalues), and a top component (eigenvectors corresponding to small number of large eigenvalues); which we call bulk-subspace and top-subspace.
>
>     A significant portion of the gradient contribution comes from the top subspace. Therefore, to understand the loss landscape and its impact on training, it is essential to focus on the stable rank, which we employ in our study.
>
> 4. **Stable rank of Hessian and Low-rank Updates**
>
>     While the Hessian may have a very high traditional rank, the number of eigenvalues contributing significantly to the loss landscape is typically small (e.g.,k-eigenvalues for k-classification problems). Stable rank effectively captures this curvature information. Stable rank is less sensitive to small perturbations in the Hessian compared to traditional rank measures ([4], [5]). This makes it a more robust metric for analyzing the training loss landscape, as minor variations in the data points or training steps do not lead to significant changes in the stable rank. This property has been widely utilized in deep learning research ([6], [7], [8]) to assess and restrict model complexity.
>
>
> Therefore, from this analysis, we expect that low-rank matrices for training restrict the complexity of local training, which leads to better performance of federated learning by reducing clients’ discrepancy

---

> > ### Comment · Reviewer_yeeh · 2024-11-23
> >
> > Thank you for the detailed reply.
> >
> > Regarding the connection between loss Hessian and weight matrices: "The connection between the Hessian and the model lies in the fact that model updates follow the eigenvectors of the Hessian."It is not clear to me why this statement is necessarily true. The algorithm performs local updates by minimizing the local objective, but I am not aware of the connection between minimization and the eigenvectors of Hessian.
> >
> > In general, I was wondering why the theory on low-rank hessians can motivate the design of low-rank weight updates. They seem very separated without further justifications.

---

> > > ### Author Response · Authors · 2024-11-25
> > > **Response to Reviewer yeeh's Comment**
> > >
> > > Thank you for the comment and raising the important question. We provide a more detailed explanation of the connection between the loss Hessian, gradient descent updates, and the motivation for employing low-rank updates in our algorithm.
> > >
> > > **Connection Between Gradient Descent and Hessian Eigenvectors**
> > >
> > > When we update the model using gradient descent, the Hessian matrix of the loss function encapsulates the local curvature information of the loss landscape. The eigenvectors of the Hessian represent the principal axes of curvature, and the corresponding eigenvalues indicate the degree of curvature along these directions.
> > >
> > > According to the several previous works ([1], [2], [3]), the direction and magnitude of the gradient $\nabla f(\omega)$ are influenced by the Hessian's eigenstructure. In particular, gradient descent updates are implicitly biased toward the eigenvectors associated with larger eigenvalues. This means that the **minimization step is dominated by the eigenvectors corresponding to the top eigenvalues**.
> > >
> > > For example, in [2], the authors projected the gradient onto the subspace spanned by the top eigenvectors and calculated the proportion of the gradient contributed by this projection. They demonstrated that the gradient is dominated by the top eigenvectors of the Hessian.
> > >
> > > **Motivation for Low-Rank Weight Updates from Low-Rank Hessian Analysis**
> > >
> > > We consider low-rank update matrices $AB$ as the **updates** and accumulate them to construct our final model. By constraining these matrices to be of low rank, we ensure that the updates lie in a low-rank subspace, aligning them with the most significant curvature directions identified by the Hessian analysis.

---

> ### Author Response · Authors · 2024-11-19
> **Continued Response to Reviewer yeeh's Review**
>
> **W1. Although the authors provide some Hessian rank structure analysis, the design of FedLoRU can be better supported from the theoretical side. For example, some convergence guarantees, since low-rank updates lead to loss of information compared to full-rank updates and may hurt the optimization.**
>
> We acknowledge that convergence analysis of LoRA (or its variants) algorithms is an important and open research area. Currently, no existing work rigorously analyzes the convergence properties of low-rank update methods, such as LoRA, in optimization. While this remains an intriguing direction for future study, our focus in this work is on demonstrating the empirical effectiveness and theoretical insights into rank-based optimization properties in federated learning.
>
> Additionally, our paper highlights that low-rank updates are particularly advantageous in federated learning environments with a large number of clients. Client discrepancies are a key factor contributing to performance degradation in federated learning, making it crucial to minimize these differences. While local training may sacrifice some client-specific information, guiding local models toward shared global knowledge is more critical in federated learning. Low-rank updates effectively achieve this by aligning local updates with the major global directions (which we call it implicit regularization effect).
>
> **References**
>
> [1] Sagun, L., Bottou, L., & LeCun, Y. (2016). Eigenvalues of the hessian in deep learning: Singularity and beyond. *arXiv preprint arXiv:1611.07476*.
>
> [2] Gur-Ari, G., Roberts, D. A., & Dyer, E. (2018). Gradient descent happens in a tiny subspace. *arXiv preprint arXiv:1812.04754*.
>
> [3] Li, T., Tan, L., Huang, Z., Tao, Q., Liu, Y., & Huang, X. (2022). Low dimensional trajectory hypothesis is true: Dnns can be trained in tiny subspaces. *IEEE Transactions on Pattern Analysis and Machine Intelligence*, *45*(3), 3411-3420.
>
> [4] Rudelson, M., & Vershynin, R. (2007). Sampling from large matrices: An approach through geometric functional analysis. *Journal of the ACM (JACM)*, *54*(4), 21-es.
>
> [5] Ipsen, I. C., & Saibaba, A. K. (2024). Stable Rank and Intrinsic Dimension of Real and Complex Matrices. *arXiv preprint arXiv:2407.21594*.
>
> [6] Bartlett, P. L., Foster, D. J., & Telgarsky, M. J. (2017). Spectrally-normalized margin bounds for neural networks. Advances in neural information processing systems, 30.
>
> [7] Neyshabur, B., Bhojanapalli, S., & Srebro, N. (2017). A pac-bayesian approach to spectrally-normalized margin bounds for neural networks. arXiv preprint arXiv:1707.09564.
>
> [8] Sanyal, A., Torr, P. H., & Dokania, P. K. (2019). Stable rank normalization for improved generalization in neural networks and gans. *arXiv preprint arXiv:1906.04659*.

---

### Official Review · Reviewer_mpJt · 2024-10-31

**Soundness:** 2
**Presentation:** 2
**Contribution:** 2
**Rating:** 3
**Confidence:** 3

**Summary:**

This paper reveals that client loss in federated learning has a higher rank structure (in gradients and Hessian subspaces) than the server's loss. Based on this, they propose that restricting client optimization to a low-rank subspace could provide implicit regularization and then introduce FedLoRU, a framework that enforces low-rank updates on the client side and aggregates them into a higher-rank model. Finally, they add another low-rank module pair to adapt to environments with statistical and model heterogeneity.

**Strengths:**

This paper reveals that client loss in federated learning has a higher rank structure (in gradients and Hessian subspaces) than the server's loss.
 Based on this, they propose that restricting client optimization to a low-rank subspace could provide implicit regularization. They then introduce FedLoRU, a framework that enforces low-rank updates on the client side and aggregates them into a higher-rank model. Finally, they add another low-rank module pair to adapt to environments with statistical and model heterogeneity.

**Weaknesses:**

The novelty is limited, there is no close connectiong between the analysis and the algorithm. I think this algorithm is a federated version of ReLoRA if we consider on the non-personalized version, aggregating low-rank modules for higher rank training.

There is no theoretical analysis for the algorithm. It's fully heuristic. When we consider the personalized strategy this paper studied, I don't know what kind of solution will this algoritm converge to. Will the introduced L, U fully concel out the A, B modules and make this algorithm fully consider local loss? The author didn't provide the reasonability of their strategy.

According to my understanding, this algorithm is still a full parameter training algorithm as it initializes W every $\tau$ step. So the comparison to LoRA is unfair. On the other hand, there are numerous algorithms for conventional federated learning. If you want to highlight your algorithm's advantage, you should compare your algorithm with the conventional algorithm, rather than just beatting LoRA.

You can't accurately solve argmin_{A,B} f. This step is computation-heavy even if you use an \epsilon-approixition. This step is actually one step of full LoRA tuning.  Therefore, this algorithm is not suitable for LLM fine-tuning.

**Questions:**

Please refer to the limitation.

---

> ### Author Response · Authors · 2024-11-19
> **Response to Reviewer mpJt's Review**
>
> We appreciate the reviewer’s time. Below, we address the concerns and questions raised:
>
> **W1. The novelty is limited, there is no close connectiong between the analysis and the algorithm.**
>
> Thank you for your feedback. We respectfully want to insist that our paper has very important technical and empirical contributions.
>
> - **Theoretical novelty 1**: This is the first work to theoretically analyze the rank nature of the optimization loss landscape in federated learning. We provide detailed analysis into the largest eigenvalues of client-side and server-side optimizations, demonstrating that local clients exhibit higher stable rank.
> - **Theoretical novelty 2**: We introduce a new approach: decouple additive perturbed models, addressing the dependency problem when analyzing the limiting eigenvalues of two Hessians. Unlike prior works such as [1], which rely on the assumption of matrix independence (which in fact, is not true: there is a dependency of Hessian of a large dataset and Hessian of a sub-dataset), our approach resolves this dependency issue, which significantly affects mathematical analysis and results.
> - **Connection between the analysis and the algorithm:** Our algorithm is directly inspired by our theoretical findings. For instance, Theorem 3.2 shows that clients have a higher stable rank, implying a more complex loss landscape of client optimization and greater client discrepancy in federated learning. By constraining client updates to low-rank representations, we align clients along major optimization directions, reducing discrepancies in training. This insight is especially impactful in settings with a large number of clients, where discrepancies naturally increase.
>
> The empirical novelty lies in our finding that client-side low-rank updates consistently outperform full-rank training in cross-device federated learning scenarios. It demonstrates the practical advantages of the proposed approach and low-rank local training.
>
> We also extend the application of low-rank training to heterogeneous settings. For pFedLoRU, unlike existing personalized federated learning methods that use full-rank models for global knowledge and low-rank models for personalized updates, we show that using low-rank models for both global and personalized training yields better performance. This is because low-rank updates for global model effectively capture general global knowledge by following major optimization directions. For mFedLoRU, we further introduce locally adaptive ranks, addressing heterogeneity across models.
>
>
> **W2. There is no theoretical analysis for the algorithm. Don't know what kind of solution will this algorithm converge to.**
>
> We would like to clarify several points regarding the theoretical analysis and the reasonability of our algorithm.
>
> Our algorithm, FedLoRU, demonstrates that: 1) It is comparable to full-rank training in federated learning (FL) in terms of performance, 2) It outperforms full-rank training in cross-device environments, where there are a large number of clients and limited participation per round.
>
> For theoretical insights regarding the rank properties and their connection to our algorithm, please refer to our response to **W1**.
>
> If the question refers to the theoretical convergence behavior of our algorithm, we acknowledge that this is an important and open research area. Currently, no existing work rigorously analyzes the convergence properties of low-rank update methods, such as LoRA, in optimization. While this remains an intriguing direction for future study, our focus in this work is on demonstrating the empirical effectiveness and theoretical insights into rank-based optimization properties in federated learning.
>
> For question about personalized strategy:
>
> In our personalized algorithm, the $AB$ module is designed to capture locally adapted knowledge, while the $LU$ module focuses on learning globally shared knowledge. The $LU$ module is shared with the global server and updated by aggregating all client contributions. To ensure proper separation of global and local knowledge, we adjust the training process. Specifically, we use more training epochs for the global module ($LU$) and train $LU$ first before updating the personalized module ($AB$). This strategy helps $LU$ learn generalized global knowledge, while $AB$ captures locally specific information.
>
> Additionally, we provide a justification for the personalized algorithm based on the rank properties observed in federated learning. By leveraging low-rank updates, we have demonstrated that they introduce an implicit regularization effect across clients. We expect that the LU module benefits from this regularization, allowing it to converge toward generalized knowledge shared by clients. Since low-rank modules are trained toward common major direction, and discrepancy between them are reduced compared to full-rank modules, we expect that the LU module encapsulates more comprehensive and general knowledge.

---

> > ### Comment · Reviewer_mpJt · 2024-11-21
> >
> > Thanks for the detailed response. The algorithm design for federated learning or personalized FL is a well-studied field. There has been a lot of well-justified work before this paper.
> >
> > This paper showed that the hessian of client loss tends to be larger than that of the server loss under certain conditions. This could be a contribution ( the bound of this difference is lacking). In addition, I don't see a connection between this algorithm and your observation. You claimed that "By constraining client updates to low-rank representations, we align clients along major optimization directions, reducing discrepancies in training". It's hard to understand why restricting the updates to low-rank updates could align with global direction. To align with global direction, many algorithms can provably attain this, such as gradient tracking employed by scaffold. Your scheme is heuristic for me.
> >
> > In addition, you keep summing these updates, which finally becomes a full-rank algorithm. I don't think your theory supports your algorithm.
> >
> > For the second point, I think the reviewer agrees with me that the theoretical analysis in this work is weak.

---

> ### Author Response · Authors · 2024-11-19
> **Continued Response to Reviewer mpJt's Review**
>
> **W3. According to my understanding, this algorithm is still a full parameter training algorithm as it initializes W every τ step. So the comparison to LoRA is unfair**
>
> We would like to clarify the nature of our algorithm and the rationale behind the comparisons presented in our work:
>
> - **Parameter Efficiency**:
>
>     Our algorithm, FedLoRU, is fully parameter-efficient training. While we reinitialize low-rank modules every τ rounds (which is optional), this does not imply that full parameters are used during training. In local training, we only train low-rank modules. In fact, one of the key advantage of FedLoRU is its communication efficiency. In federated learning, communication overhead is a critical bottleneck, and FedLoRU significantly reduces this by transferring only low-rank modules instead of a full-rank model. Therefore, comparing FedLoRU with other low-rank methods, such as FedLoRA, is both reasonable and relevant, as they share the goal of achieving communication efficiency through low-rank updates.
>
> - **Comparison with Conventional Federated Learning Algorithms**:
>
>     It is important to note that our low-rank local training strategy is highly general and can be integrated with conventional methods such as FedProx ([2]), SCAFFOLD ([3), and FedAdam ([4]). These algorithms primarily address optimization loss or server-side aggregation strategies rather than model updates. As such, we easily can plug FedLoRU algorithm to other federated learning algorithms. For example, we can use FedLoRU algorithm with FedAdam. Exploring these combinations remains an exciting avenue for future research.
>
> **W4.  You can't accurately solve argmin_{A,B} f. This step is computation-heavy even if you use an \epsilon-approixition. This step is actually one step of full LoRA tuning. Therefore, this algorithm is not suitable for LLM fine-tuning.**
>
> The one-step local training process in our algorithm is equivalent to one step of LoRA training. Numerous studies have demonstrated the effectiveness of LoRA for fine-tuning large language models (LLMs). Your statement seems LoRA itself may not be suitable for LLM fine-tuning, which contradicts existing evidence. If this interpretation is incorrect, could you please clarify your question further?
>
> If your concern is that low-rank training is unsuitable due to computational costs, I would argue the opposite—it is actually more suitable for federated LLM fine-tuning. For LLM fine-tuning, we use very low rank modules, which requires low-computational power.
>
> Especially, in the context of federated learning, the most significant bottleneck for LLM fine-tuning is not computation but communication. For instance, transferring the full weights of a model like LLaMa2-7B (~13.5GB) requires significantly more time than performing local training on client devices. By leveraging low-rank updates, our approach drastically reduces the communication overhead, making it particularly well-suited for federated LLM fine-tuning scenarios.
>
> We appreciate the reviewer's constructive comments and time again. If there is any remaining question, we will try our best to answer.
>
> **References**
>
> [1] Baskerville, N. P. (2023). Random matrix theory and the loss surfaces of neural networks. *arXiv preprint arXiv:2306.02108*.
>
> [2] Li, T., Sahu, A. K., Zaheer, M., Sanjabi, M., Talwalkar, A., & Smith, V. (2020). Federated optimization in heterogeneous networks. *Proceedings of Machine learning and systems*, *2*, 429-450.
>
> [3] Karimireddy, S. P., Kale, S., Mohri, M., Reddi, S., Stich, S., & Suresh, A. T. (2020, November). Scaffold: Stochastic controlled averaging for federated learning. In *International conference on machine learning* (pp. 5132-5143). PMLR.
>
> [4] Reddi, S., Charles, Z., Zaheer, M., Garrett, Z., Rush, K., Konečný, J., ... & McMahan, H. B. (2020). Adaptive federated optimization. *arXiv preprint arXiv:2003.00295*.

---

> > ### Comment · Reviewer_mpJt · 2024-11-21
> >
> > What I want to emphasize is that you need to do a series of LoRA, and then merge AB into W sequentially, which sacrifices the flexibility of LoRA. LoRA separating adapter and the frozen pre-trained model, which thus can be adapted to multiple tasks in parallel.  Concretely, for each task, LoRA only needs to store {A, B}. Your algorithm needs to store the parameters with the size of the full model. I think it is inefficient.
> >
> > Overall, compared with the conventional algorithm, this paper lacks theoretical justification.  Compared with LoRA, this work sacrifices flexibility and extensibility.
> >
> > I will keep my score for now.

---

> > > ### Author Response · Authors · 2024-11-25
> > > **Response to Reviewer mpJt's Comment**
> > >
> > > Thank you for your detailed comment. We address your concerns in two separate parts: (1) the connection between our theoretical analysis and the proposed algorithm, and (2) the comparison between LoRA and our algorithm.
> > >
> > > **Connection Between the Theory and the Algorithm**
> > >
> > > Our primary objective is to analyze the rank structure in federated learning and to propose an algorithmic framework that achieves better performance with a large number of clients while ensuring communication efficiency. We theoretically demonstrate that client-side optimization has a higher limiting stable rank, and we hypothesize that restricting updates to a low-rank space can align client updates with the global optimization direction. Based on this insight, we propose an algorithm that employs low-rank updates to align client updates and enhance communication efficiency.
> > >
> > > We acknowledge that you may perceive the connection between our theoretical analysis and the algorithm as not entirely direct, particularly in the hypothesis phase. To address this, we conducted experiments showing that as the number of clients increases, the performance gap between low-rank updates (FedLoRA, FedLoRU) and full-rank updates (FedAvg) widens. These empirical results support our theoretical hypothesis and demonstrate the practical effectiveness of our approach.
> > >
> > > **Comparison Between LoRA and Our Algorithm**
> > >
> > > We respectfully disagree with the assertion that our algorithm performs full-rank training. We consistently emphasize that while the updates are low-rank, the model itself is not constrained to be low-rank. We consider the low-rank update matrices $AB$ as the **updates** and accumulate them to construct the final model. By doing so, we achieve a higher-rank model using only low-rank updates. The key point is that we perform gradient descent on low-rank factorized matrices, thereby achieving the same memory and computational overhead as LoRA. Additionally, we transfer only the low-rank matrices, which results in communication efficiency.
> > >
> > > Regarding the flexibility and extensibility of LoRA, we assert that our algorithm provides the same level of adaptability. For pre-training, even if larger ranks are required for the low-rank matrices, we do not need to maintain these matrices as separate adapters since our goal is to construct a unified pre-trained model. For fine-tuning, we can retain a series of low-rank matrices separately alongside the frozen pre-trained model. Even if more low-rank matrices need to be stored, their size—especially in large language model (LLM) fine-tuning—is significantly smaller than that of the original model. This allows for a plug-and-play approach with the low-rank matrices and the pre-trained model. For example, in our LLaMa2-3B experiment, we only need to store 0.36% of the parameters, whereas FedLoRA requires storing 0.12% of the parameters.

---

> ### Comment · Reviewer_mpJt · 2024-11-25
>
> "We theoretically demonstrate that client-side optimization has a higher limiting stable rank, and we hypothesize that restricting updates to a low-rank space can align client updates with the global optimization direction." This heuristic design does not make sense to me.
>
> For the second part, I respectfully disagree with the authors.
>
> First, memory usage is definitely larger than that of LoRA. When you merge B_tA_t into W, you need extra space to store BA.
>
> Second, regarding flexibility and extensibility, "we can retain a series of low-rank matrices separately alongside the frozen pre-trained model". Actually, you merge them into the frozen model as you said in your paper. In addition, you can't claim that storing a series of BA is more efficient than storing the original model as you don't know how many of them need to be stored.

---

> > ### Author Response · Authors · 2024-12-03
> > **Response to Reviewer mpJt's Comment**
> >
> > Thank you for your detailed feedback.
> >
> > **Regarding the algorithm design:**
> >
> > Our theoretical analysis indicates that during client-side optimization in federated learning, the Hessian matrices associated with smaller local datasets tend to have a higher stable rank. This suggests that the optimization landscape at each client is high-dimensional and potentially divergent from others. By constraining the updates to a low-rank space, we aim to capture the most significant directions that are common across clients.  We acknowledge that there is a gap between our theoretical analysis and the proposed algorithm. While this does not directly conclude that constraining to a low rank will always aid in aligning client updates, our empirical results suggest otherwise (gap between full-rank training and low-rank training). We acknowledge that this heuristic may not be immediately intuitive, and we’re doing additional analysis to strengthen our hypothesis about the low-rank updates in federated learning, but due to time limit, we would be able to update it next time.
> >
> > **Regarding memory usage compared to LoRA:**
> >
> > First, you are incorrect that when merging $A_t B_t$ into $W$, additional memory is not required to store the product $B_t A_t$. The low-rank matrices can be integrated into the original model weights, resulting in no permanent increase in model size. For fine-tuning, it is correct that we need extra memory to store series of $A_t B_t$, but it is still much more efficient to use our low-rank update algorithm than using a full-rank algorithm. We agree with your point “you don't know how many of them need to be stored”, but the number of low-rank matrices stored is limited (e.g., three updates are enough for LLaMa2-3B which is less than 1% of original model).
> >
> > We recognize that this may not always be the case and will revise our manuscript to provide a more balanced discussion on this matter.
> >
> > Again, thank you for your thoughtful feedback. Your insights help us improve the clarity and accuracy of our work.

---

### Official Review · Reviewer_WVpF · 2024-11-01

**Soundness:** 2
**Presentation:** 2
**Contribution:** 3
**Rating:** 5
**Confidence:** 3

**Summary:**

To address the issue of communication efficiency and heterogeneity in Federated Learning, this paper proposes the FedLoRU method. This general low-rank update framework enforces low-rank client-side updates and accumulates these updates to form a higher-rank model. The authors provide empirical results to demonstrate that FedLoRU performs better than other algorithms.

**Strengths:**

1. The proposed method is well-motivated, the paper investigates the rank properties of client and server losses, analytically showing that under stochastic sampling, the rank of the Hessian of the loss function increases with smaller sample sizes.
2. The empirical results show empirical evidence of the higher rank structure of client losses and demonstrate that restricting the rank of local updates aids in implicit regularization.

**Weaknesses:**

1. In the theorems that are presented, summarizing the main insights of these theorems may be needed since currently they are just written as long paragraphs.
2. In experiments, the least partial client participation ratio is set as 0.5. In more realistic settings, the participation ratio is lower with more clients.
3. The author should consider more baselines, which apply low-rank factorized update models, such as [1].
[1] Nam Hyeon-Woo, Moon Ye-Bin, Tae-Hyun Oh. FedPara: Low-rank Hadamard Product for Communication-Efficient Federated Learning. ICLR 2022.

**Questions:**

See in weaknesses.

---

> ### Author Response · Authors · 2024-11-19
> **Response to Reviewer WVpF's Review**
>
> We appreciate the reviewer’s time. Below, we address the concerns and questions raised:
>
> **W1. In the theorems that are presented, summarizing the main insights of these theorems may be needed.**
>
> Thanks for the question. We will provide detailed explanation and novelty of the main theorem.
>
> - **explanation**: From a data-generating-distribution, we pick N samples, and again pick M (<N) samples from N samples. Then for prediction function h and weight w of dimension R, the stable rank of Hessian of M samples is asymptotically larger than the stable rank of Hessian of N samples.
>
> - **insight**: In federated learning with K local clients, when we assume each local client has M samples, then the server-side optimization loss is for N=KM samples. Theorem 3.2 says that when we consider the rank structure of local loss landscape and global loss landscape, local client has larger rank structure. It means that local client has more complicated loss landscape.
>
>     From this insight, we hypothesize that if we can reduce the complexity of local loss landscape (by using low-rank updates), it might help to reduce client discrepancy which is a major factor of performance degradation in federated learning. We also provide the evidence that as the local dataset size decreases (compared to the size of combined dataset), low-rank update algorithms outperform full-rank algorithm (FedAvg) (see Figure 2(b)).
>
> - **theoretical novelty 1**: This is the first theoretical analysis on rank nature of optimization loss landscape in federated learning. We provide the information about the largest eigenvalues of client-side optimization and server-side optimization, and we have shown that local clients have higher stable rank.
> - **theoretical novelty 2**: We first introduce two decoupled additive perturbed models to solve dependency problem in finding limiting eigenvalues of two Hessians. For example, Baskerville et al. just solve  the dependency problem by adding assumption that their matrices are independent, which in fact, dependent.
>
> **W2. In experiments, the least partial client participation ratio is set as 0.5. In more realistic settings, the participation ratio is lower with more clients.**
>
> Thank you for the great point.
>
> This is indeed an important consideration. Our study emphasizes the effectiveness of client-side low-rank updates, particularly in cross-device federated learning scenarios involving a large number of participating clients. To evaluate this, we conducted experiments with $K=200,300,400$, and a participation rate of $C=0.5$. These results demonstrate that low-rank training methods outperform full-rank training (FedAvg), aligning with our theoretical insights.
>
> |       $K$      | FedAvg  | FedHM  | FedLoRA | FedLoRU         |
> |--------------|---------|--------|---------|-----------------|
> | $K=100$        | 0.5382  | 0.5732 | 0.5506  | **0.5837**      |
> | $K=200$        | 0.3885  | 0.4872 | 0.5227  | **0.5393**      |
>
>
> In addition, inspired from your question, we extended our experiments to settings with a lower participation ratio and a larger number of clients. Specifically, we examined $K=100,200$ with $C=0.1$, using an IID CIFAR-100 dataset, which is more challenging than FMNIST and CIFAR-10. For these tests, we used the ResNet18 model, applying full parameter training for FedAvg and 41% parameter training for low-rank methods. The results, averaged over three runs with very low standard deviation (< 0.005), indicate that:
> - Low-rank training methods consistently outperform full-rank training under lower participation ratio with more clients.
> - FedLoRU achieves the best performance among low-rank methods.
>
> Interestingly, we observed that under these lower participation ratio conditions, FedHM surpasses FedLoRA, which contrasts with the results for higher participation ratios ($C=0.5). This finding highlights the complex relationship between participation rate and algorithm performance, further demonstrating the reliability of FedLoRU.
>
> We hope this clarifies our experimental setup and results, demonstrating the adaptability of low-rank methods in diverse participation scenarios.

---

> ### Author Response · Authors · 2024-11-19
> **Continued Response to Reviewer WVpF's Review**
>
> **W3. The author should consider more baselines, which apply low-rank factorized update models, such as [1]. (FedPara)**
>
> That is a good suggestion to compare with the other low-rank factorized updates.
>
> We agree that it is valuable to compare different low-rank methods. In our study, our primary goal was to demonstrate the advantages of low-rank training in federated learning with a large number of clients. Thus, our approach is not limited to LoRA-style low-rank updates. As mentioned in our paper (lines 348-349), other low-rank methods can also be applied, but we adopted the most standard method for our experiments.
>
> Regarding the specific method LoHa (FedPara’s low-rank approach using Hadamard products), we chose not to include it for the following reasons:
>
> 1. **Performance**: Preliminary experiments with LoHa on CIFAR-10 ($K=100$) showed that it performs worse than FedLoRA, achieving a final top-1 accuracy of approximately 0.75, which is significantly lower than other methods. Furthermore, the FedPara paper tested LoHa under settings with a small number of clients ($K=16$ for CIFAR-10 and $K=8$ for CIFAR-100) and a low participation ratio ($C=0.16$). These conditions involve only a small subset of clients participating per round, which do not align with our primary goal of testing scalability with many clients.
> 2. **Computational Overhead**: LoHa incurs significantly higher computational costs, requiring twice the training time compared to LoRA. This makes it less practical for scenarios with a large number of clients.
>
> Nonetheless, we acknowledge the importance of exploring and comparing alternative low-rank methods. Future work can investigate whether other low-rank approaches perform similarly to FedLoRU and provide a broader comparison across different methods.
>
> We appreciate the reviewer's constructive comments and time again. If there is any remaining question, we will try our best to answer.

---

### Official Review · Reviewer_9pru · 2024-11-04

**Soundness:** 2
**Presentation:** 2
**Contribution:** 2
**Rating:** 3
**Confidence:** 3

**Summary:**

The paper applies FedLoRU and its variants to impose the local update in a low-rank subspace to achieve implicit regularization.

**Strengths:**

FedLoRU uses successive low-rank updates for both pre-training and fine-tuning in federated learning and achieves good performance.

**Weaknesses:**

W1. The novelty is not justified sufficiently.

W2. More discussions and justifications regarding the stable rank metric are needed.

W3. The experiment setup and results are not convincing.

**Questions:**

Q1. The paper presents FedLoRU and its variants by applying low-rank updates in a federated learning setting. However, the novelty of this proposed method is limited. The idea of using low-rank updates in federated learning has been explored before, and the paper does not provide a compelling argument for why the proposed method outperforms existing approaches.

Q2. While the paper utilizes the stable rank metric to analyze rank properties between local clients and the central server, the discussion around this metric is lacking. The claim that stable rank "serves as a continuous proxy for rank and is robust" is made without sufficient references or supporting literature. Additionally, more discussion is needed on how this concept is adapted from related fields, and why it is appropriate for the federated learning context.

Q3. Figure 2(a) is difficult to interpret. Both the datasets with 50 and 500 samples show a high stable rank at the 15th epoch, which is counterintuitive and requires further explanation. It would strengthen the paper if the authors could repeat the experiment multiple times and provide clearer insights to support the observed trends.

Q4. The experiment shown in Figure 2(b) does not convincingly support the authors' intuition without a more detailed description. A thorough explanation of the experimental setup and its relation to Theorem 3.2 would significantly improve the clarity and impact of the results.

---

> ### Author Response · Authors · 2024-11-19
> **Response to Reviewer 9pru's Review**
>
> We appreciate the reviewer’s time. Below, we address the concerns and questions raised:
>
> **Q1. The novelty of this proposed method is limited, and the paper does not provide a compelling argument for why the proposed method outperforms existing approaches.**
>
> While we acknowledge that the concept of using LoRA in federated learning has been explored previously, our paper introduces three significant novelties that distinguish it from prior work:
>
> **[Technical Novelty] First theoretical analysis on rank nature of optimization loss landscape in federated learning.**
>
> Our work is the first to provide a theoretical analysis of the rank structure of the optimization loss landscape in federated learning. We demonstrate that local clients exhibit a higher stable rank and analyze the eigenvalues of the client-side and server-side Hessians, which offer critical insights into the curvature of the optimization landscape.
>
> These theoretical insights are essential for effectively applying low-rank updates in federated learning. Higher stable rank indicates a more complex loss landscape for clients, leading to greater client discrepancies. By constraining updates to low-rank spaces, FedLoRU mitigates these discrepancies, aligning client updates along major directions and facilitating better aggregation. Furthermore, our work introduces two decoupled additive perturbed models to address dependency issue in analyzing Hessian structure. Unlike prior approaches, such as Baskerville et al. ([1]), which assume matrix independence (which is, in real, not true), our method resolves this dependency problem, representing a significant theoretical improvement.
>
> **[Empirical Novelty] First to show low-rank updates in client-side optimization outperform full-rank training when a system has a large number of clients. We also apply client-side low-rank updates and server-side accumulation**
>
> We are the first to demonstrate that client-side low-rank updates outperform full-rank training in federated systems with a large number of clients. Our theoretical analysis suggests that low-rank updates reduce client discrepancies by simplifying the loss landscape and aligning updates along shared directions. Experimental results confirm this hypothesis, showing that even FedLoRA outperforms full-rank FedAvg. These findings position low-rank updates as a strong baseline for cross-device federated learning with large client populations. Moreover, low-rank updates are compatible with various federated learning algorithms, such as server-side optimization strategies (e.g., FedAdagrad, FedAdam [2]) and client-level frameworks like FedProx [3] and SCAFFOLD [4], further enhancing its applicability.
>
> We are also the first to apply client-side low-rank updates combined with server-side accumulation in federated learning. Inspired by the concept of ReLoRA, we adapt this idea to the federated learning setting, where the role of accumulation has not been extensively explored. Furthermore, we extend this approach to heterogeneous settings, showcasing its flexibility and potential for broader application.
>
> **Q2. More discussion is needed on how stable rank is adapted from related fields, and why it is appropriate for the federated learning context**
>
> In our paper, we use stable rank to compare the rank nature between clients and server in FL, and rank nature means the curvature information of loss landscape. In short, stable rank is the continuous proxy for rank, further, especially in deep learning, it provides more practical information about the training dynamics.
>
> - According to the works ([5], [6], [7]), gradient descent trajectory is separated into two components: a bulk component (eigenvectors corresponding to large number of small eigenvalues), and a top component (eigenvectors corresponding to small number of large eigenvalues). Furthermore, large fraction of gradient is from top eigenvectors.
>
> - Stable rank captures curvature information of loss landscape more accurately than rank. For example, Hessian would have very high rank compared to stable rank. However, even if the rank is very high, the number of eigenvalues that contributes to most of loss landscape is very small (typically, k eigenvalues for k-classification problem). Therefore, rank might not capture this curvature information accurately, but stable rank does.
>
> - Additionally, stable rank is more stable under small perturbations of Hessian ([8], [9]). Thus it is more suitable for analyzing training loss landscape. For example, small difference of the point would not affect the training landscape significantly, but the rank can be differ a lot. However, small difference of the points would not significantly change the resulting stable rank, which makes stable rank more suitable measure to capture curvature information. And this is why many deep learning studies ([10], [11], [12]) use stable rank, instead of rank, to restrict the complexity of the model.

---

> ### Author Response · Authors · 2024-11-19
> **Continued Response to Reviewer 9pru's Review**
>
> **Q3. Figure 2(a) is difficult to interpret.**
>
> Figure 2(a) illustrates the empirical stable ranks of Hessians for dataset sizes of 50 and 500, where the model is trained on the full training dataset, and stable ranks are computed during training with partial datasets. The results support Theorem 3.2, which states that the Hessian of a loss of a smaller dataset exhibits a larger stable rank at any weight when the parameter dimension is sufficiently large.
>
> The spike in stable rank observed at the 15th epoch appears to be a random phenomenon. Learning dynamics, such as the eigenvalues and stable rank of the Hessian, remain under-explored in deep learning literature. Given the inherently complex nature of the loss landscape, the stable rank may exhibit significant perturbations during training.
>
> To validate this observation, we repeated the experiments three more times. Although spikes occurred at random epochs in each trial, the consistent finding across all experiments was that smaller datasets consistently resulted in larger stable ranks except only one case (Experiment 2, r=40). This reinforces the theoretical insight provided by Theorem 3.2.
>
> | Epoch | Experiment 1 (n=50) | Experiment 1 (n=500) | Experiment 2 (n=50) | Experiment 2 (n=500) | Experiment 3 (n=50) | Experiment 3 (n=500) |
> |-------|---------------------|----------------------|---------------------|----------------------|---------------------|----------------------|
> | E=1   | 1.441               | 1.053               | 9.672               | 6.723               | 1.441               | 1.504               |
> | E=5   | 17.562              | 6.825               | 3.853               | 1.808               | 106.194             | 7.217               |
> | E=10  | 263.935             | 7.217               | 49.407              | 12.705              | 5.578               | 4.528               |
> | E=15  | 67.226              | 6.794               | 9.481               | 5.158               | 67.226              | 6.793               |
> | E=20  | 64.468              | 10.269              | 25.455              | 3.359               | 64.468              | 10.268              |
> | E=25  | 72.798              | 15.989              | 6.100               | 5.958               | 72.798              | 15.989              |
> | E=30  | 7.232               | 1.958               | 105.749             | 9.034               | 7.233               | 1.958               |
> | E=35  | 912.511             | 11.685              | 108.535             | 15.085              | 91.251              | 11.685              |
> | E=40  | 3.484               | 3.150               | 14.925              | 18.354              | 3.485               | 3.150               |
> | E=45  | 372.260             | 28.769              | 6.248               | 1.645               | 372.263             | 28.769              |
>
>
>
>
> **Q4. A thorough explanation of relation between Figure 2(b) and Theorem 3.2 improves the clarity and impact of the results.**
>
> Thank you for the suggestion. As you noted, Figure 2(b) highlights a key impact of our work.
>
> Our main contribution is demonstrating the benefits of client-side low-rank optimization in federated learning with many clients. We compared full-rank training (FedAvg) with low-rank approaches (FedLoRU, FedLoRA) under the same framework, differing only in local updates. Figure 2(b) shows that as client numbers increase, low-rank methods outperform full-rank training. The performance gap between FedAvg and FedLoRU grows with more clients, and even FedLoRA surpasses FedAvg for $K \in {200, 300, 400}$.
>
> This is especially relevant for cross-device federated learning, where many edge devices participate. Our findings suggest that low-rank updates are more effective than full-rank training in such settings.

---

> ### Author Response · Authors · 2024-11-19
> **References for the Responses**
>
> We appreciate the reviewer's constructive comments and time again. If there is any remaining question, we will try our best to answer.
>
> We provide the references for the responses.
>
> **References**
>
> [1] Baskerville, N. P. (2023). Random matrix theory and the loss surfaces of neural networks. arXiv preprint arXiv:2306.02108.
>
> [2] Li, T., Sahu, A. K., Zaheer, M., Sanjabi, M., Talwalkar, A., & Smith, V. (2020). Federated optimization in heterogeneous networks. *Proceedings of Machine learning and systems*, *2*, 429-450.
>
> [3] Reddi, S., Charles, Z., Zaheer, M., Garrett, Z., Rush, K., Konečný, J., ... & McMahan, H. B. (2020). Adaptive federated optimization. *arXiv preprint arXiv:2003.00295*.
>
> [4] Karimireddy, S. P., Kale, S., Mohri, M., Reddi, S., Stich, S., & Suresh, A. T. (2020, November). Scaffold: Stochastic controlled averaging for federated learning. In *International conference on machine learning* (pp. 5132-5143). PMLR.
>
> [5] Sagun, L., Bottou, L., & LeCun, Y. (2016). Eigenvalues of the hessian in deep learning: Singularity and beyond. *arXiv preprint arXiv:1611.07476*.
>
> [6] Gur-Ari, G., Roberts, D. A., & Dyer, E. (2018). Gradient descent happens in a tiny subspace. *arXiv preprint arXiv:1812.04754*.
>
> [7] Li, T., Tan, L., Huang, Z., Tao, Q., Liu, Y., & Huang, X. (2022). Low dimensional trajectory hypothesis is true: Dnns can be trained in tiny subspaces. *IEEE Transactions on Pattern Analysis and Machine Intelligence*, *45*(3), 3411-3420.
>
> [8] Rudelson, M., & Vershynin, R. (2007). Sampling from large matrices: An approach through geometric functional analysis. *Journal of the ACM (JACM)*, *54*(4), 21-es.
>
> [9] Ipsen, I. C., & Saibaba, A. K. (2024). Stable Rank and Intrinsic Dimension of Real and Complex Matrices. *arXiv preprint arXiv:2407.21594*.
>
> [10] Bartlett, P. L., Foster, D. J., & Telgarsky, M. J. (2017). Spectrally-normalized margin bounds for neural networks. Advances in neural information processing systems, 30.
>
> [11] Neyshabur, B., Bhojanapalli, S., & Srebro, N. (2017). A pac-bayesian approach to spectrally-normalized margin bounds for neural networks. arXiv preprint arXiv:1707.09564.
>
> [12] Sanyal, A., Torr, P. H., & Dokania, P. K. (2019). Stable rank normalization for improved generalization in neural networks and gans. *arXiv preprint arXiv:1906.04659*.

---

> ### Comment · Reviewer_9pru · 2024-11-25
>
> The theoretical analysis reveals a higher-rank nature of Hessian of a smaller dataset, but it may not be regarded as a conclusion for federated learning. The insight only reveals the high-rank nature of small datasets, and it can not conclude constraining to low rank would help to align client updates along major directions and facilitate better aggregation. Therefore the contribution of the theory is limited. Also as the algorithmic convergence wasn't established, the conclusion is not convincing without repeating your experiment to report the variance.

---

> > ### Author Response · Authors · 2024-12-03
> > **Response to Reviewer 9pru's Comment**
> >
> > Thank you for your feedback. We acknowledge that there is a gap between our theoretical analysis and the proposed algorithm. Our theoretical work highlights the high-rank nature of the Hessian in local optimization loss, which we believe provides valuable insights into the optimization landscape of federated learning. While this does not directly conclude that constraining to a low rank will always aid in aligning client updates, our empirical results suggest otherwise.
> >
> > Specifically, our experiments demonstrate that as the number of clients increases, the performance gap between the low-rank algorithm and the full-rank algorithm widens. This empirical observation indicates that low-rank constraints can indeed help in aligning client updates along major directions, facilitating better aggregation in federated settings with many clients. Further, we’re doing additional analysis to strengthen our hypothesis about the low-rank updates in federated learning, but due to time limit, we would be able to update it next time.
> >
> > We also have taken care to report variance across multiple runs to ensure the reliability of our findings. In fact, we have low variance across multiple runs for each setting. We appreciate your insights and will consider them to improve the clarity and impact of our work.

---

### Meta-Review · Area_Chair_4nMH · 2024-12-21

**Metareview:**

The paper proposes the FedLoRU method: a federated optimization method combining low-rank updates on the clients' side (equivalent to running LORA locally on each client), aggregating the updates on the servers, and repeating this process. The authors claim this method is motivated by the needs for communication-efficient federated learning methods, combined with regularization offered by low-rank adaptation.

I do not really observe any particular strengths in this paper. The core of the idea is straightforward, and appeared in some previous papers already (e.g., Kuo et al, 2024). The difference here is that multiple low-rank updates are added to the base model over the algorithm run. However, this idea was explored before, too, in the COLA (Chain of LORA) paper (Xia et al, 2024).

Some issues:

- There is no convergence analysis of the new method.
- The aggregation mechanism is problematic from a theoretical point of view: the A updates are aggregated on their own, and the B updates are aggregated on their own, whereas the "mathematically correct" update would be to average the products A*B; i.e., average the updates. This approach does not lead to a low-rank update, however. Thus, the authors resort to a heuristic, which needs justification. Multiple papers were written on this topic before.
- The connection between weights being low-rank and Hessians being of low-rank, which plays a key role in their justification of the method, was questions by the reviewers, and the explanation is not satisfying - it relies on hand-waving arguments instead of on solid mathematical reasoning.
- Experimental results were not found to be convincing enough - a very high bar is needed for an empirical work; and this bar was certainly not achieved.
- The authors seem to be not aware of prior FL literature with very closely connected algorithms which, unlike their work, have strong theoretical backing. For example, literature on federated optimization with contractive compression applied to the updates by the clients (e.g., all work on error feedback, started in 2014 by Seide, with hundreds of follow-up works) is very closely connected. This is because low-rank approximation is known to be a contractive compressor.
- I doubt the convergence of the presented method can be analyzed mathematically - in fact, I believe simple counterexamples can be found on which this methods fails.

Finally, no reviewer recommended this paper for acceptance. I've read the reviews, the rebuttals and the discussion, and have briefly looked at the paper as well. I agree with the overall judgement that this paper has many significant weaknesses, and should not be accepted.

AC

**Additional Comments On Reviewer Discussion:**

The key here is that the points raised by the reviewers were not addressed satisfactorily. This was also not possible, since the issues are indeed issues with the paper, and not merely due to misunderstanding by the reviewers that can be explained away.

---

### Decision · Program_Chairs · 2025-01-22

Reject